# Transforming biorefinery designs with 'Plug-In Processes of Lignin' to enable economic waste valorization

Zhi-Hua Liu[1,2], Naijia Hao[3], Yun-Yan Wang [3], Chang Dou[4], Furong Lin[1,2], Rongchun Shen[5], Renata Bura[4], David B. Hodge[6], Bruce E. Dale[7], Arthur J. Ragauskas [3,8,9], Bin Yang[5] & Joshua S. Yuan [1,2✉]

Biological lignin valorization has emerged as a major solution for sustainable and cost-effective biorefineries. However, current biorefineries yield lignin with inadequate fractionation for bioconversion, yet substantial changes of these biorefinery designs to focus on lignin could jeopardize carbohydrate efficiency and increase capital costs. We resolve the dilemma by designing 'plug-in processes of lignin' with the integration of leading pretreatment technologies. Substantial improvement of lignin bioconversion and synergistic enhancement of carbohydrate processing are achieved by solubilizing lignin via lowering molecular weight and increasing hydrophilic groups, addressing the dilemma of lignin- or carbohydrate-first scenarios. The plug-in processes of lignin could enable minimum polyhydroxyalkanoate selling price at as low as $6.18/kg. The results highlight the potential to achieve commercial production of polyhydroxyalkanoates as a co-product of cellulosic ethanol. Here, we show that the plug-in processes of lignin could transform biorefinery design toward sustainability by promoting carbon efficiency and optimizing the total capital cost.

[1] Synthetic and Systems Biology Innovation Hub, Texas A&M University, College Station, TX, USA. [2] Department of Plant Pathology and Microbiology, Texas A&M University, College Station, TX, USA. [3] Department of Chemical & Biomolecular Engineering, University of Tennessee, Knoxville, TN, USA. [4] School of Environmental and Forest Sciences, University of Washington, Seattle, WA, USA. [5] Bioproducts, Sciences, and Engineering Laboratory, Department of Biological Systems Engineering, Washington State University, Richland, WA, USA. [6] Chemical and Biological Engineering Department, Montana State University, Bozeman, MT, USA. [7] Biomass Conversion Research Laboratory, Department of Chemical Engineering and Materials Science, Michigan State University, East Lansing, MI, USA. [8] Biosciences Division, Oak Ridge National Laboratory, Oak Ridge, TN, USA. [9] Department of Forestry, Wildlife and Fisheries, Center for Renewable Carbon, The University of Tennessee Institute of Agriculture, Knoxville, TN, USA. ✉email: syuan@tamu.edu

Lignocellulosic biorefineries are critical to empower the emerging bioeconomy and to advance energy and environmental sustainability[1–3]. The recent consensus is that a sustainable biorefinery heavily depends on deconstructing and valorizing three major components (cellulose, hemicellulose, and lignin) of lignocellulosic biomass (LCB) simultaneously[4–7]. As the second most abundant terrestrial polymer after cellulose, lignin has a significant potential to serve as a sustainable source for the production of fuels, chemicals, and materials[8–11]. The conversion of lignin into fungible products thus could substantially improve the sustainability of renewable biofuels[12–15]. However, the limitation of current biorefinery lies in the focus on exploiting more uniform carbohydrates for biofuels, while leaving the lignin as an underutilized biorefinery waste[16–19].

Recently, lignin bioconversion has been shown to have significant promise towards its valorization, as some microorganisms in nature have evolved metabolic pathways to convert heterogeneous aromatics via 'biological funnel' to valuable products[20–23]. Among the different microbes, ligninolytic *Pseudomonas putida* have an extensive toolbox for metabolizing lignin oligomers and aromatics to synthetize polyhydroxyalkanoates (PHAs), which is an intracellular carbon and energy reserve compounds[24–28]. The conversion of biorefinery wastes to PHAs could improve environmental sustainability from two aspects. On one hand, as a class of biodegradable polyesters, PHAs could be applied for the production of renewable bioplastics, addressing the daunting environmental challenges caused by petrochemical plastics[29–31]. On the other hand, the valorization of lignin to PHAs could make a biorefinery more economic and sustainable[32–34]. Current biorefinery scenarios have been built for rendering more processible carbohydrate, while the chemical structure of lignin is not optimal for bioconversion, considering the inefficient fractionation, the poor lignin processibility, and the low yield of target products. Most importantly, current pretreatment employed high severities, which could generate more condensed lignin and hamper the bioconversion efficiency[4,35,36]. The concept of lignin-first biorefinery was proposed recently, where the biorefinery design will consider how to best tailor lignin chemistry and generate less condensed structure for downstream processing[16,37,38]. However, lignin-first fractionation preserves more β-O-4 linkages and produces high lignin molecules, which is not suitable for bioconversion[16,38]. The lignin-first scenario also needed to exploit new fractionation technologies and could require substantial changes of current pretreatment design and potential compromise of carbohydrate processing for fuels. The ideal situation is to avoid substantial changes of current pretreatment and biorefinery design, yet achieving efficient lignin utilization. However, this is a very challenging task, as the current pretreatment and biorefinery were not designed to render processible lignin, but a waste stream for low-cost combustion.

These leading pretreatment technologies cannot overcome the intrinsic nature of lignin and yield a lignin stream suitable for bioconversion. In general, pretreatments catalyzed by acid, steam explosion, and liquid hot water typically break the β-O-4 linkages between lignin units to render smaller lignin segments[39,40]. However, these pretreatments exploited a high-temperature, causing lignin to coalesce into larger molten bodies that migrate within and out of the cell wall[41,42]. They may also increase the guaiacyl and condensed phenolic units, and hence promote the repolymerization of lignin to macromolecules with condensed structure[39]. Ammonia fiber explosion (AFEX) is to be capable of cleaving ester and lignin-carbohydrate bonds, such as *p*-coumarate esters and ferulate-polysaccharide esters, but it preserves the overall chemical structure of the lignin polymer and retains most of the lignin in biomass[43,44]. For these reasons, the yield and processibility of lignin are still inadequate to enable the bioconversion at a high titer due to its low solubility and reactivity. To address these technical challenges, advanced processing technologies are required to alter lignin chemistry and improve its solubility and reactivity. In particular, it is critical to systemically evaluate how processible the lignin derived from current pretreatments are, what type of chemical structures could enable better lignin processibility, and how to achieve efficient lignin bioconversion without compromising carbohydrate-based biofuel production. It is also necessary to evaluate how the lignin valorization can be processed in current biorefinery and what extent the incorporation of lignin valorization could contribute to the overall output and the capital cost of biorefinery.

In this study, we address these challenges with the design of 'plug-in processes of lignin (PIPOL)' for biological lignin valorization that can be directly incorporated into current biorefinery. The research systemically evaluates lignin bioconversion performance on the integration of PIPOL with five leading pretreatments in biorefinery. The PIPOL has been designed with the integration of solubilization, conditioning, and fermentation to improve the solubility and reactivity of lignin and its microbial conversion in biorefinery designs. The performance of the PIPOL is evaluated by comparing the yields, stream characteristics and processibility of lignin, and the titer and yield of PHAs produced by engineered *P. putida* KT2440. The characteristics of the lignin stream are then carried out to illustrate the chemical mechanisms of the improved processibility for microbial conversion. Finally, the techno-economic analysis (TEA) has been conducted to assess the performance of biological lignin valorization and its contributions to the overall profitability of biorefinery.

## Results

**'Plug-in processes of lignin' improves lignin dissolution.** The fundamental challenges for the bioconversion of biorefinery waste lies in the fact that most of the current pretreatment and hydrolysis platforms generate a largely solid lignin waste stream, which is not amenable to bioconversion. Our previous studies have established that lignin dissolution is critical for bioconversion, as aromatic monomers and oligomers can serve as substrates for microbes, but not the larger molecules[21,45–47]. Based on this understanding, we have designed a set of procedures, dubbed as 'plug-in processes of lignin (PIPOL)', to integrate the solubilization, conditioning, and fermentation for lignin bioconversion. PIPOL is incorporated with current biorefinery to achieve lignin dissolution and bioconversion (Fig. 1a–f). Considering the features of different leading pretreatments, the PIPOL designs were implemented with various biorefinery configurations and compared in terms of lignin conversion, PHA yield, saccharification efficiency, and economics.

In detail, five leading pretreatments that have the potential to be commercially implicated were integrated with PIPOL to maximally unleash their roles in improving deconstruction efficiency, achieving lignin processibility, and enhancing biorefinery performance (Supplementary Table 1). As the soluble low-molecular-weight lignin will enable its microbial conversion[24,48], the solubilization of the pretreated solid was developed to tailor lignin chemistry and enhance its biological processibility. The soluble lignin was more amenable to microbial conversion into PHA after conditioning. The processing steps of solubilization, conditioning, and fermentation for lignin bioconversion were designed as 'PIPOL' to be incorporated into a biorefinery. For dilute sulfuric acid (DSA) and liquid hot water (LHW) pretreatments (Fig. 1b, d), the lignin solubilization was conducted after separating the pretreated solid in the pretreatment. The soluble lignin stream from the solubilization can be mixed well with the liquid stream from pretreatment to eliminate the conditioning of lignin stream. This PIPOL design integrated the

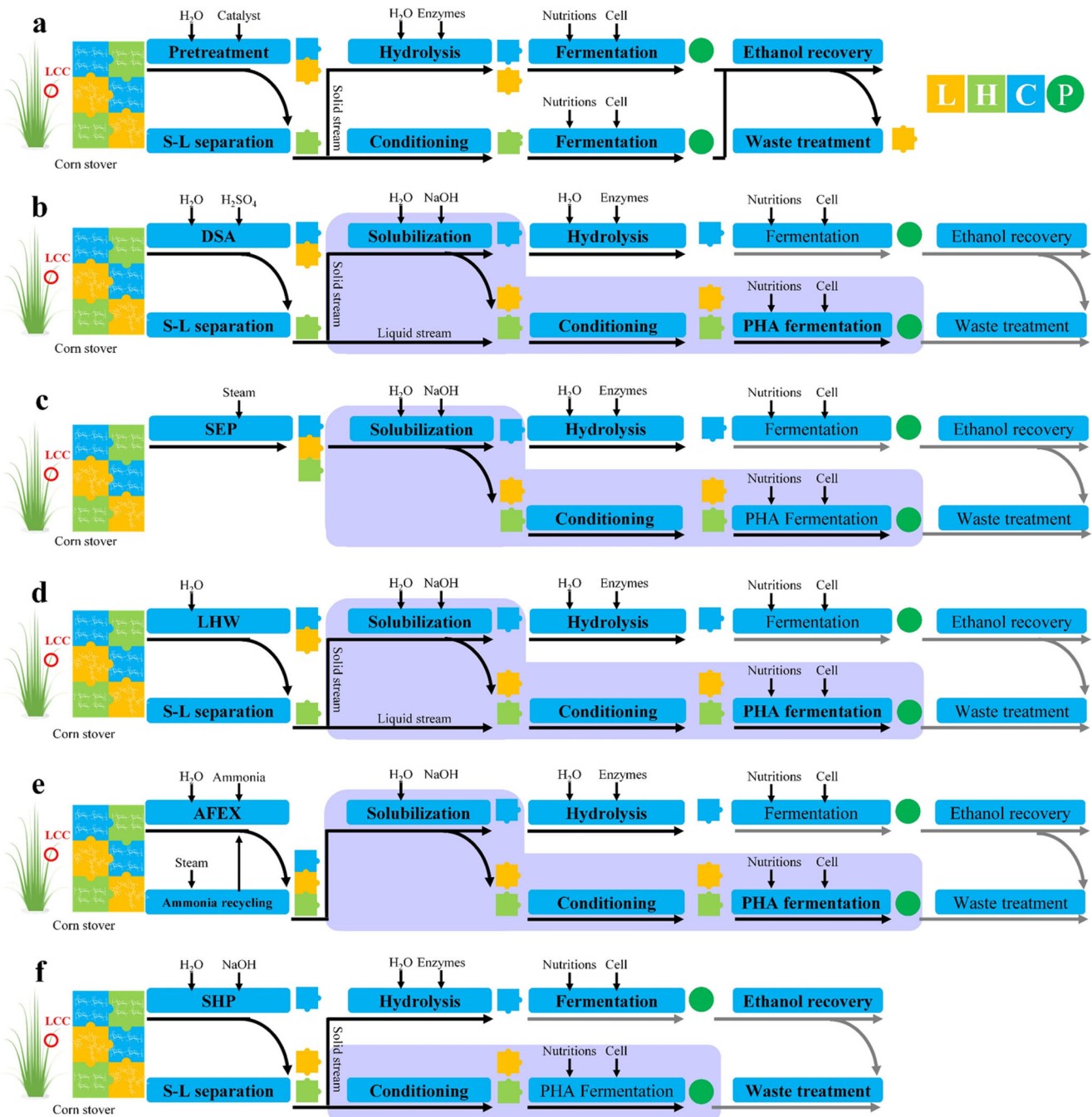

**Fig. 1 The biorefinery scenarios with the incorporation of 'plug-in processes of lignin (PIPOL)'. a** A general biorefinery process for bioethanol production; **b** PIPOL-integrated biorefinery scenario with dilute sulfuric acid pretreatment (DSA); **c** PIPOL-integrated biorefinery scenario with steam explosion pretreatment (SEP); **d** PIPOL-integrated biorefinery scenario with liquid hot water pretreatment (LHW); **e** PIPOL-integrated biorefinery scenario with ammonia fiber expansion (AFEX); **f** PIPOL-integrated biorefinery scenario with sodium hydroxide pretreatment (SHP). Purple module represents PIPOL by integrating the dissolution, conditioning, and bioconversion of lignin. L in orange square represents lignin, H in light green square represents hemicellulose, C in blue square represents cellulose, P in green circle represents product, S–L separation represents solid–liquid separation, LCC represents lignin–carbohydrate complex.

solubilization and fermentation for lignin bioconversion in a biorefinery, where carbohydrates were enzymatically hydrolyzed after the lignin solubilization. As steam explosion pretreatment (SEP) and ammonia fiber expansion (AFEX) were carried out at high solids, the PIPOL design integrated the solubilization, conditioning, and fermentation following the pretreatment (Fig. 1c, e). Sodium hydroxide pretreatment (SHP) was employed to directly fractionate the lignin from LCB for microbial conversion (Fig. 1f).

The lignin–carbohydrate complex (LCC) and lignin content are the primary contributors to the recalcitrance of plant cell wall[17,49]. Monitoring component transformation could provide insight into how each pretreatment and solubilization may change these components. It could also help to define to what degree the biorefinery designs overcame the recalcitrant nature of biomass to improve the accessibility of sugars and lignin[50,51]. As expected, five pretreatments exhibited distinct impacts on biomass chemistries, demonstrating by differences in cell wall

components of pretreated corn stover. As shown in Supplementary Fig. 1, all pretreatments except AFEX resulted in significant changes in cell wall components compared with corn stover feedstock (Supplementary Table 2). DSA, SEP, LHW, and SHP enriched glucan content in solid fraction and each showed specific impact on the content of xylan and lignin. One observation was that the vast majority of the xylan is removed from the solid fraction by DSA and SEP and dissolved into the liquid phase. The acidic pretreatments enable the xylan solubilization via the autohydrolysis effect because acetyl groups linked to hemicelluloses are hydrolyzed to form acetic acid to catalyze the xylan depolymerization[52,53]. The results clearly showed that the majority of the lignin and part of the xylan were solubilized in the liquid phase by SHP, which can be recovered and ultimately used as carbon sources in the subsequent fermentation. Alkaline pretreatment generally cleaved ester and glycosidic side chains to solubilize lignin, remove acetyl group, and partially solubilize hemicellulose[36,54,55]. AFEX did not obviously affect the content of glucan, xylan, and lignin because AFEX is a 'dry-to-dry' pretreatment and does not remove any component from biomass[56,57].

After pretreatment, PIPOL design further deconstructed the pretreated solid, altered the component content, and solubilized lignin. The component transformation showed significant differences among the pretreated solids after the PIPOL. The lignin content after the PIPOL decreased by 39.1, 38.2, 40.0, and 61.2% compared with that in the DSA-, SEP-, LHW-, and AFEX- treated solid, respectively, indicating the enhanced lignin dissolution. As the PIPOL could partially dissolve hemicellulose in the liquid phase, the xylan content in the solids after PIPOL could result from a balance of the removal of lignin and xylan. Correspondingly, due to the solubilization of lignin and the removal of xylan, the glucan content after PIPOL increased by 24.5, 36.1, 25.2, and 84.1% compared with that in the DSA-, SEP-, LHW-, and AFEX-treated solid, respectively. AFEX enriched glucan significantly after the solubilization due to the removal of lignin and other decomposition products generated in AFEX[43]. Generally, the high glucan content of the solids will benefit biorefinery by increasing target product concentration and reducing its separation cost.

The depolymerization of lignin to low-molecular-weight derivatives is potentially an efficient route to facilitate the microbial conversion for valuable products[21,48,58,59]. As the bioconversion efficiency depended on the yield of water soluble lignin, the lignin transformation was monitored in the PIPOL-integrated biorefinery (Fig. 2). First, PIPOL significantly improved lignin dissolution, when integrated with all pretreatment conditions. Most pretreatments will yield lignin-rich solid streams by releasing the carbohydrate after hydrolysis, which will impede the bioconversion due to high molecular weight and low reactivity of these lignins. PIPOL yielded the soluble lignin of 70.2, 65.6, 56.2, 81.5, and 48.7% from DSA-, SEP-, LHW-, AFEX-, and SHP-treated solids, respectively. Second, the dissolving behavior of lignin largely depended on the pretreatment and the PIPOL employed. The fractionation of AFEX, DSA, and SEP followed by PIPOL possessed a high proportion of dissolved lignin, potentially promoting the bioconversion of the soluble lignin. Overall, pretreatments have their unique role in impacting the component transformation to overcome biomass recalcitrance, and the PIPOL actually further enriched the glucan in the solid and concentrated the soluble lignin in the liquid phase, which may facilitate the utilization of these components in the subsequent conversion process.

improve lignin bioconversion without compromising carbohydrate utilization. PIPOL actually improved hydrolysis efficiency for all of the leading pretreatments. Pretreatments have their unique role in altering structural properties of LCB, such as contents of lignin, hemicellulose, and acetyl group, cellulose structure, particle size etc. These changes will increase the accessibility of carbohydrates and hence the hydrolysis performance[60–62]. As expected, all pretreatments promoted the conversion of glucan (Fig. 3a, b) and xylan (Fig. 3c, d). The initial conversion of glucan and xylan was determined by the accessibility of carbohydrates and can be used to assess the performance of pretreatment and PIPOL[61,63]. The results showed that SHP- and AFEX- treated solids produced higher initial conversion of glucan and xylan as compared to SEP-, DSA-, and LHW-treated ones (Supplementary Fig. 2). Generally, lignin represents one of the most important factors limiting the rate and extent of hydrolysis because it acts as a physical barrier to restrict the accessibility of carbohydrates to enzymes[17,60,64]. The results suggested that the accessibility of carbohydrates was significantly improved by AFEX likely due to the cleavage and redeposition of lignin polymer within the cell walls and by SHP possibly due to the removal of lignin and xylan fraction[36,43,60]. After 168 h of hydrolysis, AFEX and SHP also led to higher glucan and xylan conversion as compared with DSA, SEP, and LHW.

Interestingly, the PIPOL further improved the enzymatic hydrolysis performance. Such increases varied significantly among different pretreatments. The PIPOL produced 3.1, 3.3, and 3.6% per hour of the initial glucan conversion rate for the DSA-, SEP-, and AFEX-treated solids, respectively, while it increased the initial glucan conversion rate by 71.7, 56.7, and 50.4% (Supplementary Fig. 2). Meanwhile, the PIPOL increased the initial xylan conversion rate for all pretreated solids. Results implied that the PIPOL can further improve the accessibility of carbohydrates due to the removal of lignin, xylan, and other components and thereby increased the initial sugar conversion rate. After 168 h hydrolysis, the glucan conversion after the PIPOL reached 89.7, 94.9, 80.1, 96.8, and 84.2% for DSA-, SEP-, LHW-, AFEX-, and SHP-treated solids, respectively (Fig. 3b). The PIPOL incorporating with SEP and AFEX facilitated the improvements of enzymatic hydrolysis.

High solid enzymatic hydrolysis shows the potential to be used in industrial implementation of lignocellulosic ethanol as it provides higher sugar concentration and requires lower operating cost[61,65,66]. PIPOL improved the high solid enzymatic hydrolysis performance. Figure 3e, f shows that the trends of glucan and xylan conversions at high solids were similar to those at low solids. The glucan and xylan conversion at high solids were slightly lower than those at low solids likely due to the water constraint and the limitation of mass transfer[61,65]. However, the PIPOL increased glucan conversion by 13.3, 11.8, 21.6, and 10.6% for the DSA-, SEP-, LHW-, and AFEX-treated solids, respectively.

The PIPOL therefore significantly improved the sugar release from corn stover biomass (Supplementary Fig. 3). The PIPOL following AFEX produced the highest yield of both glucose and xylose as compared to other pretreatments[50,67]. The sugar yield increases largely followed the order of the PIPOL with AFEX, SEP, DSA, SHP, and LHW. Overall, the biorefinery designs integrating the pretreatment with PIPOL improved sugar conversion and yield. Such improvement was achieved for all leading pretreatment technologies used in this study, whereas the integration of PIPOL with SEP and AFEX led to the most significant improvement of enzymatic hydrolysis and sugar release.

**PIPOL facilitates the enzymatic hydrolysis of carbohydrates.** One of the key goals of PIPOL-integrated biorefinery design is to

**PIPOL promotes microbial lignin conversion to produce PHAs.** Here, we use a ligninolytic organism, engineered *P. putida*

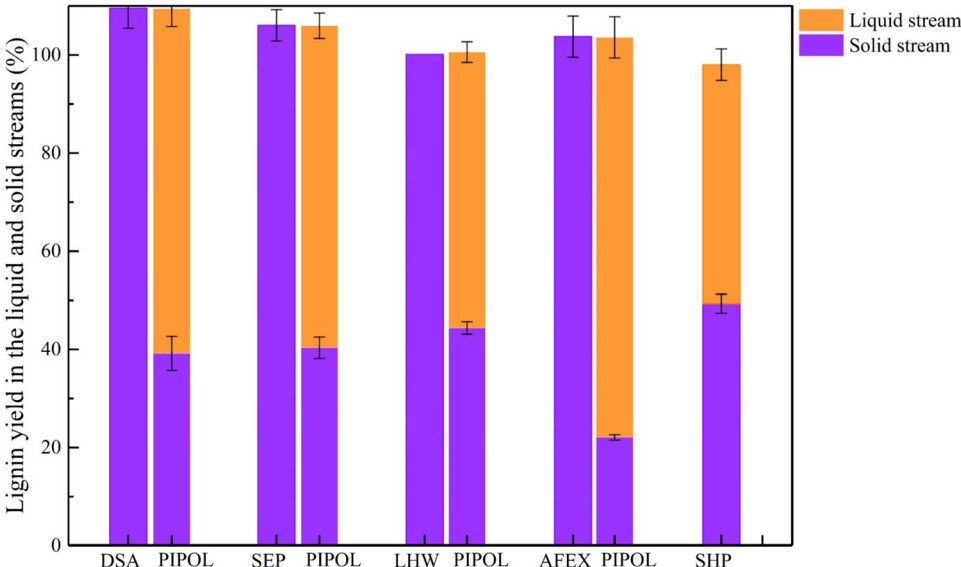

**Fig. 2 The lignin yield in the solid and liquid streams by 'plug-in processes of lignin (PIPOL)'.** DSA dilute sulfuric acid pretreatment, SEP steam explosion pretreatment, LHW liquid hot water pretreatment, AFEX ammonia fiber expansion, SHP sodium hydroxide pretreatment. The conditions of leading pretreatment were provided in Supplementary Table 1. Error bars represent the standard deviation.

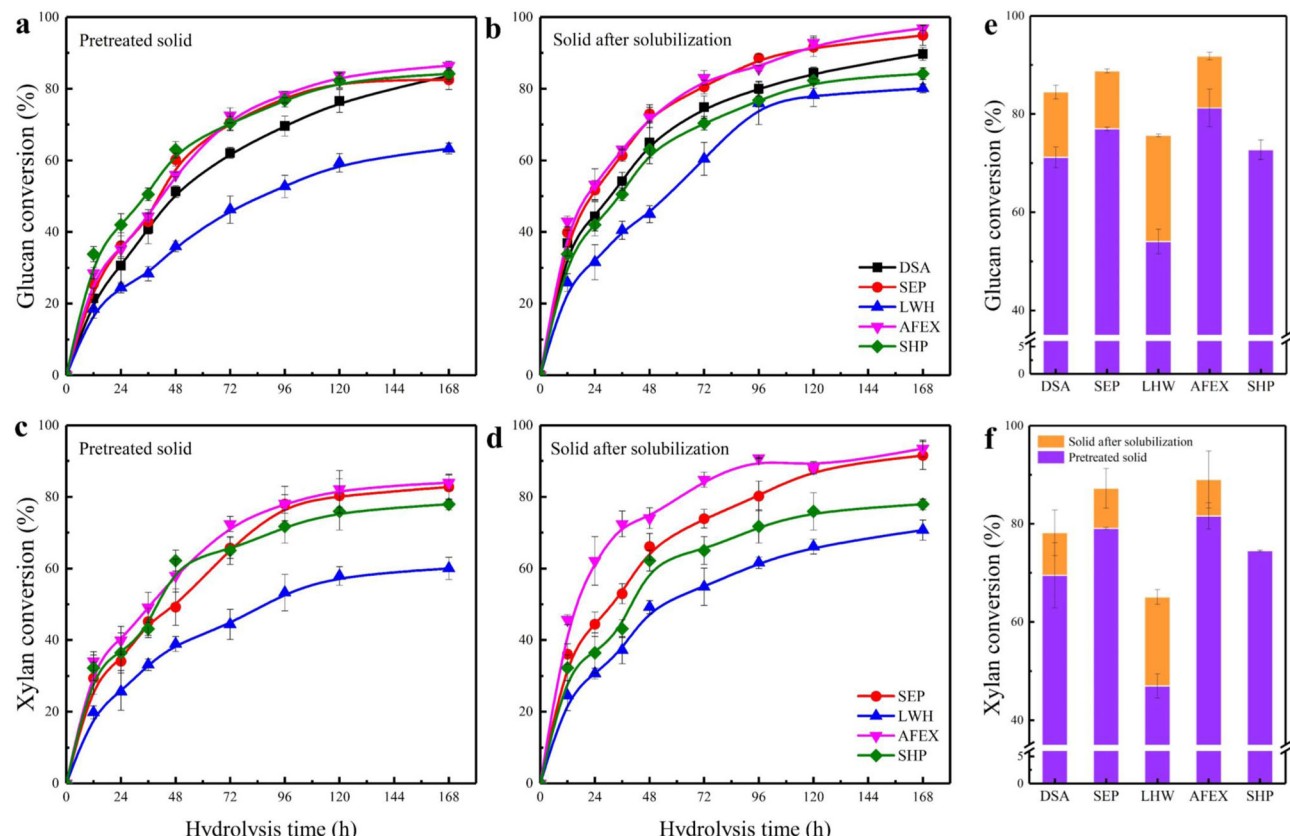

**Fig. 3 Enzymatic hydrolysis performance of the carbohydrate solid streams.** Kinetics of enzymatic hydrolysis was conducted at 1% solid loading for 168 h (**a–d**). **a** and **b** represents the glucan conversion of the solids produced from pretreatment and solubilization, respectively, while **c** and **d** represents the xylan conversion of the solids produced from pretreatment and solubilization. High solid enzymatic hydrolysis was conducted at 10% solid loading (**e**, **f**). **e** and **f** represents the glucan and xylan conversion at high solid enzymatic hydrolysis, respectively. DSA dilute sulfuric acid pretreatment, SEP steam explosion pretreatment, LHW liquid hot water pretreatment, AFEX ammonia fiber expansion, SHP sodium hydroxide pretreatment. Error bars represent the standard deviation.

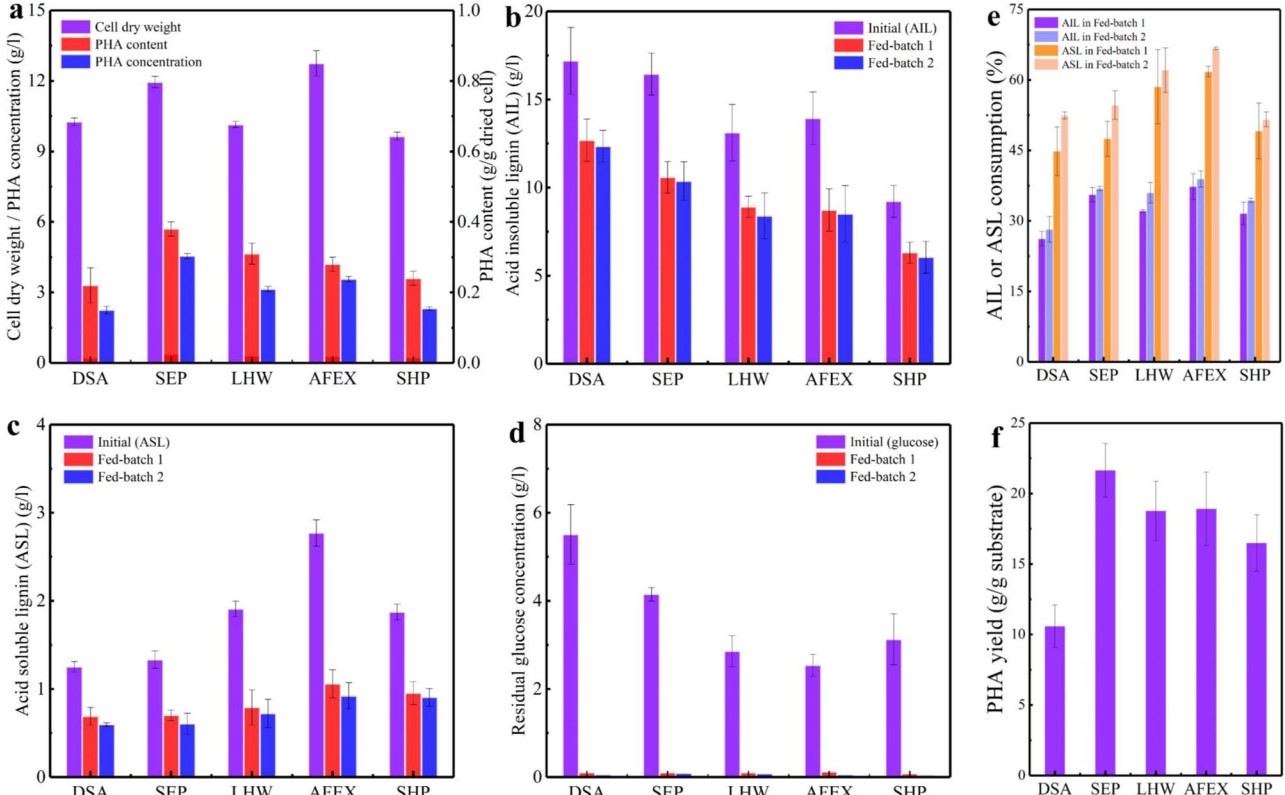

**Fig. 4 The microbial conversion of the soluble lignin to polyhydroxyalkanoates (PHAs). a** It represents cell dry weight, PHA concentration, and PHA content in fed-batch fermentation using different soluble lignins as carbon sources; **b, c** It represents the concentration of the acid insoluble and acid soluble lignin, respectively; **d** It represents the concentration of residual sugar; **e** It represents the lignin consumption in fed-batch fermentation; **f** It represents PHA yield based on the consumption of substrates. AIL acid insoluble lignin, ASL acid soluble lignin, DSA dilute sulfuric acid pretreatment, SEP steam explosion pretreatment, LHW liquid hot water pretreatment, AFEX ammonia fiber expansion, SHP sodium hydroxide pretreatment. Initial represents the initial concentration (acid insoluble lignin, acid soluble lignin, and residual glucose). Error bars represent the standard deviation.

KT2440, to demonstrate that the soluble lignin from the PIPOL can be used to synthesize the PHAs. Microbial conversion performance can be promoted with the improved processibility of lignin (Fig. 4). The soluble lignin stream has a high pH value, and the exact pH depended on the pretreatment employed (Supplementary Fig. 4). Generally, a neutralization step was needed to adjust pH to facilitate the bacteria growth. After conditioning to an optimal pH ~7.0, the soluble lignin stream was evaluated for PHA fermentation. Interestingly, even though the lignin was obtained from the same feedstock, the biological processibility significantly depended on the fractionation process employed. Figure 4a shows that cell dry weight after fed-batch fermentation reached 12.7 g/l and 11.9 g/l using the soluble lignin produced from the PIPOL with AFEX and SEP, respectively, indicating the improved cell growth with better processibility of lignin. PHA content in dried cell increased according to the order of DSA < SHP < AFEX < LHW < SEP, corresponding to 0.22, 0.24, 0.28, 0.31, and 0.38 g/g dried cell, respectively. The soluble lignin produced from the PIPOL with SEP, LHW, and AFEX facilitated the accumulation of PHAs in *P. putida* KT2440. The PHA concentration thus reached to 2.2, 2.3, 3.2, 3.6, and 4.5 g/l, respectively, using the soluble lignin produced from the PIPOL with DSA, SHP, LHW, AFEX, and SEP. The soluble lignin from SHP has been shown to have reasonable processibility for bioconversion[24]. The PHA concentration from AFEX and SEP was 1.6 and 2.0 times as that from SHP, respectively, indicating the improved bioconversion of lignin. Figure 4b, c, e shows that *P. putida* KT2440 consumed more soluble lignin produced from the PIPOL with AFEX, LHW, and SEP than DSA and SHP.

The solubilization in PIPOL produced the soluble lignin stream and generated some residual glucose (Fig. 4d). As the residual glucose concentration in the soluble lignin stream is too low to be separated and utilized alone, it was co-processed to improve the lignin bioconversion, improving the overall carbohydrate utilization. Although DSA and SHP produced more residual glucose to facilitate the PHA fermentation, AFEX and SEP led to better cell growth, higher PHA concentration and yield (Fig. 4f), and thus better fermentation efficiency. Taken together, among different pretreatments, AFEX and SEP had the best compatibility with PIPOL to produce more processible lignin. The PIPOL design thus has its unique advantages to improve the biological processibility of lignin and promote its conversion to PHA.

**The chemical mechanisms for enhanced lignin processibility.** The differential biological processibilities indicated that lignin produced from each pretreatment and PIPOL could have its unique chemical properties and bioreactivity. To understand structure-processibility relationship, the lignin properties were characterized by different analytical technologies.

The lignin molecular weight represents a fundamental property determining its microbial conversion performance. Results showed that the lignin molecular weight depended on the pretreatment and PIPOL employed (Supplementary Fig. 5a, b, c). DSA, SEP, and LHW exhibited hardly any changes of the lignin molecular weight compared with corn stover native lignin (CSNL) possibly due to the competition between depolymerization and repolymerization of lignin during acidic

pretreatment[68–70]. However, AFEX and SHP decreased the weight-average molecular weight of lignin by 69% and 43% and the number-average molecular weight by 60% and 33%, respectively, suggesting obvious depolymerization of lignin. As the lignin with low molecular weight could have high solubility to facilitate the bioconversion, the PIPOL was carried out to further deconstruct the lignin polymer in the pretreated solids. As expected, the PIPOL produced more lower-molecular-weight lignin. After the PIPOL, the highest decreases in weight-average molecular weight and number-average molecular weight were 81% and 76%, respectively, which were obtained from SEP-treated lignin.

The results of fermentation confirmed that low-molecular lignin could be consumed by *P. putida* KT2440, which was indicated by the increase of weight-average molecular weight of lignin (Supplementary Fig. 5d, e, f). When the soluble lignin from the PIPOL with SEP and AFEX was used, more than a 40% increase in the weight-average molecular weight was observed after fermentation, which was consistent with better PHA fermentation performance. Therefore, the PIPOL effectively depolymerized the lignin to low-molecular-weight derivatives, promoted its consumption, and thus facilitated its bioconversion. Such an effect is particularly efficient for SEP and AFEX, which renders lignin with highest cell biomass yield for bioconversion.

The PIPOL with leading pretreatments especially SEP and AFEX decreased the amount of β-O-4 and β-5 linkages in the soluble lignin as compared to those in CSNL (Supplementary Fig. 6), indicating efficient lignin depolymerization. The improved lignin depolymerization enhanced lignin dissolution. Besides, the PIPOL with SEP and AFEX were more enriched with H- and G- type lignin, which could also facilitate lignin bioconversion, as *P. putida* KT2440 possess more effective metabolism pathway of these aromatics[71].

The functional hydrophilic groups of lignin including hydroxyl and carboxyl groups not only could reflect the degree of depolymerization, but also could impact on the lignin solubility and thus reactivity in microbial conversion. The changes in the functional groups of the lignin from each pretreatment and PIPOL were analyzed by $^{31}$P NMR (Supplementary Figs. 7 and 8). It was observed that the functional groups of lignin depended on the biorefinery design. The content of the aliphatic hydroxyl group decreased by 18–67% after pretreatment as compared with CSNL, which was consistent with the previous studies[45,72,73]. Acidic pretreatments of DSA, SEP, and LHW significantly increased total phenolic hydroxyl and carboxyl groups as compared with AFEX and SHP. The decreased aliphatic hydroxyl group is attributed to the dehydration reaction, while the increased phenolic hydroxyl group mainly resulted from the cleavage of β-O-4 linkages.

Interestingly, the PIPOL further decreased the content of the aliphatic hydroxyl group and increased the total phenolic hydroxyl and carboxyl groups of the soluble lignin in general (Supplementary Figs. 7 and 8), indicating the enhanced depolymerization and the improved solubility of the lignin. During fermentation, these functional groups showed different behavior affecting on the lignin reactivity. The content of aliphatic hydroxyl and phenolic hydroxyl groups actually decreased after fermentation, possibly due to the efficient metabolization of lignin molecules with high solubility by *P. putida* KT2440. However, it should be noted that the C5 substituted hydroxyl group contributed the most to the phenolic hydroxyl group of the lignin produced from the PIPOL with DSA and SHP, which could be correlated with the generation of condensed lignin[35,64,74]. The results helped to explain why less lignin was consumed and low PHA was accumulated in fermentation for these two scenarios. Using the

soluble lignin from the PIPOL with SEP and AFEX, the total phenolic hydroxyl groups decreased by around 20% and 80% after fermentation, respectively, which correlated to the better bioconversion performance.

Overall, the PIPOL efficiently depolymerized the lignin by breaking more β-O-4 and β-5 linkages and thus produced lower molecular weight lignin with more hydrophilic groups, as compared to conventional pretreatment alone. SEP and AFEX by themselves led to more efficient lignin depolymerization, and the integration of PIPOL further increased lignin solubility and accessibility. The results highlighted the molecular mechanisms for better lignin processibility in bioconversion by PIPOL and showed how such impact is different among various pretreatments and biorefinery designs.

**Techno-economic analysis of biological lignin valorization.** Techno-economic analysis (TEA) of the biological lignin valorization needs to be thoroughly investigated to compare the feasibility of integrating the PIPOL into current biorefineries and elucidate the commercial relevance and technical barriers for the PIPOL[75–77]. TEA could also help to identify the primary cost drivers that empower the profitability of lignin valorization and forecast the economic impact of the biorefinery through process design and optimization, mitigating the technical risk for scaling-up. TEA of the bioconversion of lignin to PHAs has thus been carried out to assess the PIPOL efficiency by determining a minimum PHA selling price (MPSP) and identifying the primary cost drivers.

The PIPOL for lignin valorization has been designed and projected to be integrated into the biorefinery plant. All other capital costs for the overall biorefinery were assumed to be identical to those for the pretreatment developed based on NREL 2011[76,78]. In the TEA, a discounted cash flow calculation has been employed to calculate the MPSP, which meets a 10% after-tax internal rate of return. The economics of the lignin bioconversion to PHAs is summarized for the scenarios with different PIPOL technologies from the initial techno-economic evaluation (Table 1). Results indicated that the calculated MPSP will come down as the technologies developed in this project mature over the life of the project. It can be observed that the MPSP for the biorefinery scenario 5 employed SHP-PIPOL was $11.99/kg, which was highest among these scenarios. Notably, the SHP-PIPOL exhibited the lowest lignin yield, PHA yield, and PHA titer through microbial conversion. The MPSP for AFEX-PIPOL and SEP-PIPOL was $6.18/kg and $6.82/kg, respectively, which was lower than other biorefinery designs. These two biorefinery designs also showed higher lignin yield, PHA yield, and PHA titer compared with others. The variation tendency of MPSP coincides with that of annual PHA production from lignin streams, showing that PHA yield and titer are the most crucial factors affecting the MPSP, although the total capital cost and the total operating cost varies with every other factor as well.

A further analysis of the unit cost revealed the key cost contributors of MPSP in the PIPOL, when integrated with lignocellulosic biorefinery (Fig. 5a–e). Results showed that the contributors of the MPSP varied with the different biorefinery designs employed. The operating costs (47–58%), including the utilities, maintenance materials, operating supplies, operating labor, maintenance labor, and control lab labor, etc. were found to be the largest cost contributor per unit of PHAs produced. As one of the major cost contributors in large-scale PHA production, the raw material cost (31–44%) was the second largest cost contributor for the MPSP[77,78]. Besides, the annual fixed capital cost was found the third highest cost contributor, following taxes and insurance in the present study.

**Table 1 Techno-economic analysis (TEA) of the biological lignin valorization to polyhydroxyalkanoates (PHAs).**

|  | Scenario 1 DSA-PIPOL | Scenario 2 SEP-PIPOL | Scenario 3 LHW-PIPOL | Scenario 4 AFEX-PIPOL | Scenario 5 SHP-PIPOL |
|---|---|---|---|---|---|
| Annual production/MMkg | 2.19 | 3.21 | 2.6 | 3.54 | 2.02 |
| Total capital cost/MM$ | 44.98 | 42.64 | 42.13 | 39.85 | 42.12 |
| Total operation cost/MM$/yr | 11.2 | 12.42 | 12.26 | 13.24 | 14.65 |
| Raw material/MM$/yr | 4.88 | 5.9 | 5.77 | 7.4 | 8.12 |
| Utilities/MM$/yr | 0.268 | 0.267 | 0.266 | 0.270 | 0.266 |
| Unit cost/$/kg | 7.52 | 5.49 | 6.73 | 5.05 | 9.9 |
| Rate of return/% | 10 | 10 | 10 | 10 | 10 |
| Minimum selling price/$/kg | 9.58 | 6.82 | 8.35 | 6.18 | 11.99 |

Scenarios 1–5 represent the biorefinery integrated 'plug-in processes of lignin (PIPOL)' with dilute sulfuric acid pretreatment (DSA), steam explosion pretreatment (SEP), liquid hot water pretreatment (LHW), ammonia fiber expansion (AFEX), sodium hydroxide pretreatment (SHP), respectively.

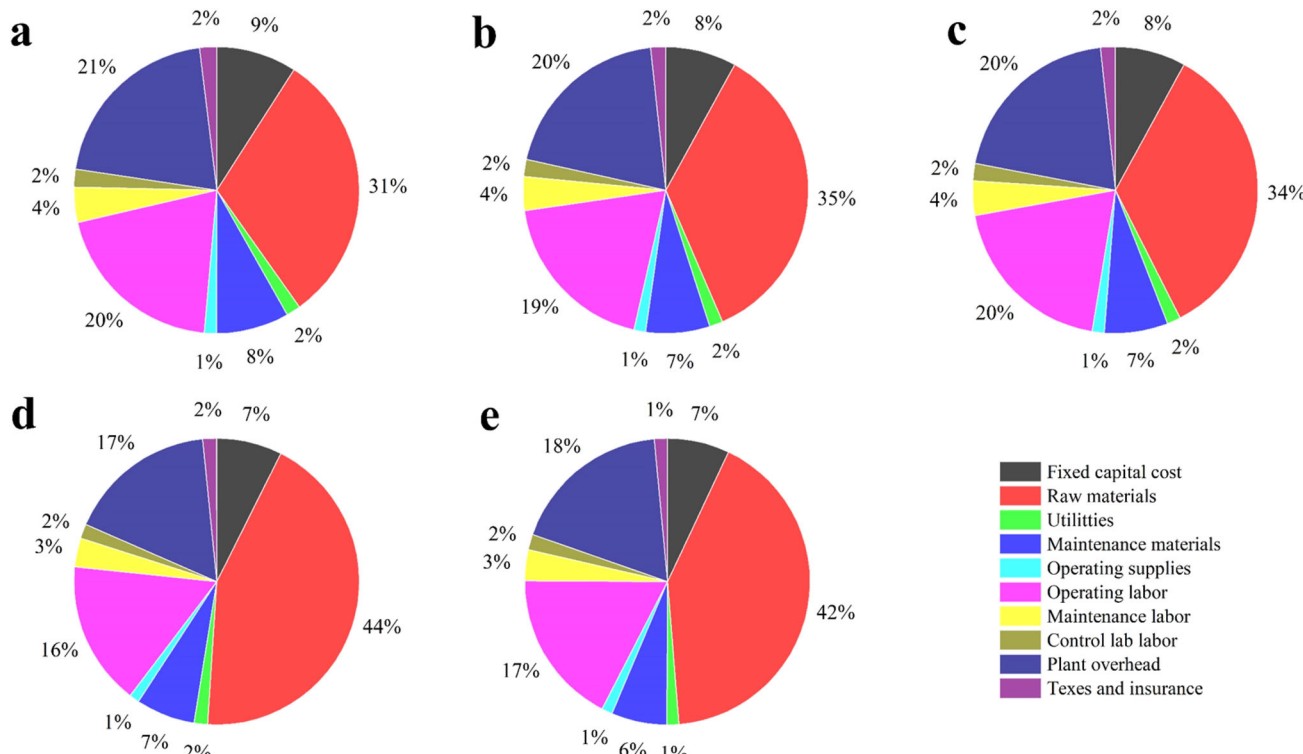

**Fig. 5 Cost contribution of minimum polyhydroxyalkanoate selling price (MPSP). a–e** It represents biorefinery scenario 1–5 (Table 1), which was integrated "plug-in processes of lignin (PIPOL)' with dilute sulfuric acid pretreatment (DSA), steam explosion pretreatment (SEP), liquid hot water pretreatment (LHW), ammonia fiber expansion (AFEX), sodium hydroxide pretreatment (SHP), respectively.

## Discussion

The PIPOL-based biorefinery addressed the dilemma between carbohydrate- or lignin-first biorefinery. Lignin is a promising feedstock to generate a range of biofuels, chemicals, and materials, which is critical to accelerating the industrial implementation of biorefinery. Carbohydrate-first biorefinery employed various pretreatment technologies to release the carbohydrate, leaving the lignin as the solid wastes. The classic pretreatments and biorefinery designs were not optimized for lignin bioprocessing. To enable efficient lignin utilization, lignin-first biorefinery have emerged and been built with lignin valorization as the primary targets[16,37]. This scenario could prevent lignin repolymerization by employing stabilization strategies and thus facilitate the selective lignin utilization[16,38]. However, the isolated lignin yield may be limited and in turn negatively affects the profitability of the lignin upgrading. The additional units and complicated

operations were required to build an independent process for lignin upgrading, which would increase the overall capital cost. The process built for lignin-first scenario could be also at the expense of carbohydrate utilization and thus weaken the output and yield of biofuel from lignocellulosic biorefinery.

In order to address the dilemma of carbohydrate- or lignin-first biorefinery, a strategy was hereby designed to incorporate the lignin valorization into current biorefinery. There are two main concerns when this strategy is implemented to facilitate microbial lignin conversion in a biorefinery. First, we need to understand how much lignin the fractionation design could solubilize and what type of chemistry the design can introduce to improve the processibility and the subsequent bioconversion of lignin. Second, we need to understand how the lignin bioconversion can be incorporated into current biorefinery to facilitate the simultaneous valorization of three main components, and cannot be

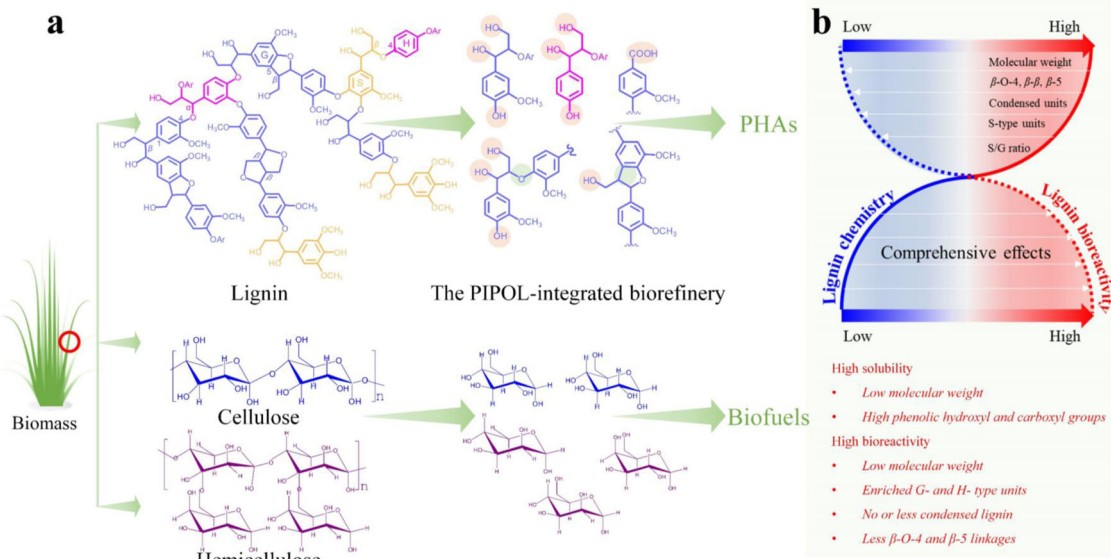

**Fig. 6 The proposed mechanisms for the improved biological lignin valorization in biorefinery. a** PIPOL-integrated biorefinery deconstructed lignin–carbohydrate complex of lignocellulosic biomass to enhance lignin solubility and carbohydrate accessibility; **b** The potential relationships between lignin chemistry and bioreactivity for microbial conversion. PIPOL represents plug-in processes of lignin. PHAs represent polyhydroxyalkanoates; S-, G-, and H-type represents syringyl, guaiacyl, and *p*-coumaryl unit of lignin, respectively.

performed at the expense of the effective utilization of carbohydrates.

The study well addresses these questions with biological lignin valorization designed as a 'plug-in process of lignin (PIPOL)' to be incorporated in a biorefinery (Fig. 1). This PIPOL integrated the solubilization, conditioning, and fermentation for biological valorization of lignin streams. In particular, five leading pretreatments with commercial potential were integrated with PIPOL to improve both lignin processibility and carbohydrate hydrolysis performance. The results highlighted that the soluble low-molecular-weight lignin was produced with tailored chemistry and enhanced processibility to facilitate the microbial conversion. The soluble lignin was valorized to PHAs by fermentation to achieve 4.5 g/L PHA at maximum. Moreover, PIPOL also synergized with promoted carbohydrate hydrolysis. The biorefinery with PIPOL thus represented an effective scenario to address the dilemma of lignin- or carbohydrate-first biorefinery, by simultaneously improving the processing efficiency of lignin and the hydrolysis performance of carbohydrates.

PIPOL represents a universal solution, while different leading pretreatments have various potentials for lignin valorization with PIPOL. The results highlighted that PIPOL has significantly improved both lignin dissolution and bioconversion in all five leading pretreatments (Fig. 2 and Supplementary Fig. 9). Simultaneously, PIPOL effectively improved hydrolysis efficiency of carbohydrate in all pretreatments. The principle and technology can be broadly applied to produce different bioproducts including munoic acid, adipic acid, lipids, and such from lignin stream. The real application will be dependent on the yield of a particular product and the economics.

Despite its broad applicability, different PIPOL configurations thus showed variations in their improvements of lignin and carbohydrate processibilities (Fig. 6 and Supplementary Fig. 10). When PIPOL is integrated with SEP and AFEX, the biorefinery design rendered the most improved lignin dissolution, bioconversion product yield, and enzymatic hydrolysis of carbohydrate (Fig. 6a). The superior performance is due to their super capacity to deconstruct the LCC of biomass. AFEX exhibits the potentiality to dissolve and extract lignin by ammonolysis, and

redeposit the dissolved lignin and decomposition products on the surface of pretreated cell walls, when the ammonia evaporates[43,44]. AFEX also causes cellulose swelling and transforms the crystal structure from cellulose I to cellulose III[44]. Followed by AFEX, PIPOL further removed the deposited lignin and the decomposition products to form nanoporous tunnel-like networks within the cell walls. These modifications from AFEX followed by PIPOL improved the accessibility of carbohydrates and thereby improved the hydrolysis performance significantly. PIPOL with AFEX effectively broken down β-O-4 and β-5 linkages, depolymerized the lignin to produce the low-molecular-weight derivatives, yielded lignin fraction carried more hydrophilic hydroxyl and carboxyl groups, enriched H- and G- type lignin, and thus promoted its consumption and PHA synthesis. As for SEP, it exhibits the capacity to dissolve hemicellulose, and solubilize and redeposit lignin from biomass by auto-hydrolysis effects[79]. It can explode the biomass particles into small pieces, disrupt the crystalline cellulose, and form the pore within biomass by instantaneous decompression[53,79]. Followed by SEP, PIPOL removed the deposited lignin and partly dissolved the xylan to improve the accessibility of carbohydrates. PIPOL further catalyzed the lignin and yielded the lower molecular lignin with more total hydrophilic groups for improving the processibility. For DSA, it removed the hemicellulose from the biomass solid and may promote the lignin phase transition, causing lignin to coalesce into larger molten bodies and redeposit on the surface of plant cell walls[60,80]. Followed by DSA, PIPOL removed the deposited lignin on the surface or within the cell walls by the disruption of ester and glycosidic side chains to improve the enzymatic hydrolysis.

PIPOL-integrated biorefinery designs have the potential to transform lignocellulosic biorefinery and biofuel industry with better economics and sustainability. First, PIPOL significantly promoted the microbial conversion of lignin to PHAs through improving the yield, the solubility, and the reactivity of lignin. The PIPOL integrated with SEP and AFEX produced a soluble lignin yield of 65.6% and 81.5%, respectively. These scenarios also depolymerized the lignin to low-molecular-weight derivatives and functionalized the lignin with more hydrophilic groups, enhancing the solubility and reactivity. The comprehensive effects of

more soluble and reactive lignin have led to the significantly improved processability for ligninolytic bacteria (Fig. 6b and Supplementary Fig. 10). In fed-batch fermentation, the PHA concentration from the PIPOL integrated with SEP and AFEX reached a comparable level of 3.6 and 4.5 g/l, respectively. The improved lignin bioconversion to valuable PHAs provided unique opportunities to enhance the cost-effectiveness and sustainability of biorefinery and to bring about significant energy and environment benefits due to the conversion of waste stream into valuable products.

Second, PIPOL enabled the synergistic improvements of carbohydrate accessibility and hydrolyzability. The PIPOL-integrated pretreatment improved biomass deconstruction, altered its structure and component, and thus promoted the enzymatic conversion of carbohydrates. PIPOL increased initial glucan conversion by 56% and 50% from SEP- and AFEX-treated solids, respectively, while it enabled a final glucan and xylan conversion of more than 95% and 92%. The PIPOL-integrated biorefinery also increased sugar yield and had the potential to reduce the enzyme loading, and improve ethanol fermentation performance. Further optimization could lead to the optimized biorefinery with less enzyme inputs and maximized ethanol output, along with value-added lignin-based products.

Third, the initial economic analysis revealed significant potential of the lignin valorization. The PIPOL-integrated biorefinery employed SEP and AFEX generated the MPSP of $6.82/kg and $6.18/kg, respectively. PIPOL minimized the costs of lignin upgrading by improving the yield of both lignin and PHAs, which are the most crucial factors affecting the MPSP. TEA results confirmed that PIPOL minimized the requirement of the additional units, equipment, and other complicated operations for lignin processing, and thus reduced the total capital cost of microbial conversion to PHAs[77,81,82]. Meanwhile, the PIPOL-integrated biorefinery could maximize the synergistic utilization of carbohydrates and lignin, improve the product outputs, amortize the capital investment and hereby facilitate the industrial implementation of biorefineries.

TEA further highlighted the advantage of PIPOL-integrated biorefinery as compared to lignin- or carbohydrate-first scenarios. The integrated process simplifies the operations by directly cooperating into a biorefinery plant such as lignocellulosic ethanol plant. TEA results suggested that higher yields and titers for the lignin bioconversion into valuable coproducts such as PHAs were essential to improve the profitability of lignin valorization. The technology development guided by TEA will produce the soluble lignin with specific bioreactivity and reduce the operating cost and the raw material cost. Taken together, the projected MPSP will be further decreased with the development of the mature technologies and the advent of biorefineries.

Overall, the PIPOL-integrated biorefinery designs synergistically improved the biological processing of lignin and the enzymatic accessibility of carbohydrate, maximized the overall output of products, and minimized the costs of lignin upgrading. The innovation of PIPOL has significant potential opportunity to make biorefinery profitable and sustainable with future development.

## Methods
**Pretreatment strategies of corn stover biomass**. Five leading pretreatments were employed to deconstruct the corn stover and facilitate the fractionization of soluble lignin for microbial conversion (Supplementary Table 1). Dilute sulfuric acid pretreatment (DSA) was conducted with a holding temperature of 150 °C and a residence time of 10 min at 10% (w/w) solid loading and 0.2% $H_2SO_4$ (w/w) in 2.0-L stainless steel Parr reactor systems (Parr Instruments Company, USA) (Supplementary Fig. 11). After pretreatment, the slurry was filtered by vacuum filtration to separate solid from liquid stream. These samples were collected for further use. Steam explosion pretreatment (SEP) was carried out in a 2.7-L batch steam explosion reactor chamber (Aurora Technical, Savona, BC, Canada). Corn stover was pre-soaked in deionized water with a liquid to solid ratio of 5:1 for 30 min to increase the moisture and then centrifuged to obtain a moisture content of 60%. The corn stover was then treated at 200 °C for 10 min[83]. Liquid hot water (LWH) pretreatment was performed in a 5-L M/K Systems digester (M/K Systems, Inc., USA). Corn stover was loaded into the cylinder and the pretreatment was programmed to heat up to 160 °C at a heat rate of 0.8 °C/min, holding for 5 min, followed by cool-down for 1 h at a rate of ~1 °C/min. Ammonia fiber expansion (AFEX) of corn stover was conducted at 140 °C for 30 min with ammonia loading of 1.0 g/g dry biomass[57,84,85]. The pretreated corn stover was collected, filled into sealed plastic bags, and stored at −4 °C for further use. Sodium hydroxide pretreatment (SHP) was conducted at 10% (w/w) solid loading and 1% NaOH (w/w) at 150 °C and for 10 min. After pretreatment, the slurry was filtered to separate solid from liquid stream. After conditioning, the liquid stream containing soluble lignin was used as carbon source in PHA fermentation.

**Lignin solubilization of the pretreated corn stover**. For lignin bioconversion, the solubilization of the pretreated solid was conducted to produce soluble lignin and to improve lignin reactivity for microbial conversion. In detail, 50 g pretreated solid (dw) was loaded into a 1.0-L screw neck bottle with a polypropylene pour ring at 10% (w/w) solid loading with 1% NaOH (w/w). To avoid the significant degradation of sugars and minimize the generation of condensed lignin, the solubilization was conducted with mild conditions of 121 °C for 30 min using Amsco LG 250 Laboratory Steam Sterilizer (Steris, USA). The heating time was about 5 min, while the cooling time was about 10 min. The slurry was vacuum filtrated through a Brinell funnel to separate the soluble lignin stream from the solid residues. After conditioning, the liquid stream containing soluble lignin was used as carbon source in fermentation. The operations of solubilization, conditioning, and fermentation were designed to improve lignin solubility and reactivity for microbial conversion (Fig. 1). As these three steps were directly incorporated into current biorefinery to achieve co-utilization of lignin and carbohydrate at high efficiency. They were named as 'Plug-In Processes of Lignin (PIPOL)'.

**Enzymatic hydrolysis**. The enzymatic hydrolysis assays were conducted using the enzyme preparations of Cellic CTec2 and HTec2. Filter paper activity (FPU) of Cellic CTec2 is 96 FPU ml⁻¹, and the cellobiase activity of β-glucosidase is 1270 CBU ml⁻¹. To evaluate the performance of pretreatment and solubilization, the enzymatic hydrolysis of the solid fraction was performed at 1% (w/w) solid loading in a 0.05 M citrate buffer solution (pH 4.8) with a Cellic CTec2 loading of 5 FPU/g glucan and a volumetric ratio 9:1 of CTec2 and HTec2 in a 250-ml Erlenmeyer flask at 50 °C and 200 rpm for 168 h. A 1.0 ml sampling of supernatant was collected at 12, 24, 36, 48, 72, 96, 120, and 168 h for sugar analysis. Sugar conversion was calculated based on the solid used in the hydrolysis. Initial hydrolysis rate was calculated at the first 12 h hydrolysis.

High solid enzymatic hydrolysis was conducted at a solid loading of 10% (w/w) with a Cellic CTec2 loading of 5 FPU/g glucan and a volumetric ratio 9:1 of CTec2 and HTec2 under 50 °C and 200 rpm for 168 h. After the hydrolysis, the hydrolysate was centrifuged at 13,800 × g for 5 min (Avanti JXN-30, Beckman Coulter, USA). The solid residues were washed with a volume of water equal to 20 times the dry weight of the solid. The liquid was collected for sugar analysis. All these experiments were conducted with two replicates.

**Polyhydroxyalkanoate (PHA) fermentation**. The soluble lignin stream from the solubilization was used as carbon source to produce PHAs by engineered *P. putida* KT2440. The seed culture followed the procedure shown in Supplementary Information[48]. *P. putida* KT2440 cell pellets for inoculation were collected by centrifuging the seed culture at 2200 × g for 10 min (Avanti JXN-30, Beckman Coulter, USA). To prepare medium, the soluble lignin was adjusted carefully to pH 7.0 by 1.0 M sulfuric acid, and mixed well with 10 ml 10X Basal salts and 1 ml 100X Mg/Ca/B1/Goodies mixture to make 100 ml medium. Fermentation was conducted using fed-batch mode in a 250-ml Erlenmeyer flask with a working volume of 100 ml at pH 7.0, 28 °C, and 200 rpm for 24 h.

For the analysis of cell dry weight and PHA content, the cell biomass was harvested by centrifugation at 13,800 × g for 10 min (Avanti JXN-30, Beckman Coulter, USA) and washed two times using 0.9% NaCl solution. The cell biomass was freeze-dried by a lyophilizer at −50 °C for 24 h (Labconco Corporation, USA). Cell dry weight was determined by a gravimetric method. PHA content in dried cell was analyzed using the GC–MS method. In detail, a 5–10 mg amount of freeze-dried cells was subjected to methanolysis with a solution containing 2 ml of 15% (v/v) sulfuric acid in methanol and 2 ml chloroform at 100 °C for 140 min. After cooling, 1 ml of ddH2O was added and the lower chloroform organic phase containing the resulting methyl esters was separated at 1200 × g for 10 min (Avanti JXN-30, Beckman Coulter, USA). A total of 0.5 ml chloroform mixture and 0.5 ml 0.1% caprylic acid in chloroform were mixed well and filtered with 0.22-μm polytetrafluorethylene (PTFE) membrane. Gas chromatography–mass spectrometry (GC–MS) analysis was performed on a GC-MS-QP2010SE (Shimadzu Scientific Instruments, Inc.) with a Shimadzu SH-Rxi-5Sil column (30 m × 250 μm × 0.25 μm). The eluted sample was analyzed using helium as a carrier gas at a flow rate of 1.0 ml/min, while the temperature maintained at 50 °C

for 3 min and then increased to 300 °C with 10 °C/min. The injector and transfer line temperature were maintained at 250 °C. Mass spectral peak quantification was performed using GCMSsolution software Ver. 2.6. All experiments were performed in triplicate.

**Lignin characterization.** For the preparation of native lignin from corn stover, the air-dried corn stover was milled for 2 h at 600 rpm using a planetary ball mill (Retsch PM 100) with zirconium dioxide vessels (50 ml) containing $ZrO_2$ ball bearings (10 mm × 10). The milled samples were then enzymatically hydrolyzed using Cellic CTec2 (0.1 ml/g solid) and HTec2 (0.1 ml/g solid) at 50 °C for 24 h. The supernatants were removed by centrifugation, and the solid residues were hydrolyzed again under the same conditions. The residue was then extracted with 96% dioxane at ambient temperature for 48 h[64,86]. The corn stover native lignin (CSNL) was recovered using a rotary evaporator and then freeze-dried for further analysis. 2D $^{1}$H-$^{13}$C HSQC nuclear magnetic resonance (NMR) and $^{31}$P NMR spectra of the lignin samples was obtained to analyze the depolymerization and modification of the lignin and evaluate its processibility for bioconversion[87–89]. Gel-permeation chromatography (GPC) was employed to determine the molecular weight of the lignin samples[8]. The analysis procedures of NMR and GPC employed in this study were shown in Supplementary Information.

**Techno-economic analysis (TEA) methodology.** Techno-economic models include a conceptual level of process design to develop a detailed process flow diagram, rigorous materials and energy balance calculation, capital and project cost estimation, a discounted cash flow economic model, and the calculation of a MPSP[77,90–93]. Aspen Process Economic Analyzer 10.1 was employed to estimate free-on-board equipment costs. Peters and Timmerhaus investment factors were used to calculate project capital expenditures[94]. Five pretreatment processes for the lignin liberation from corn stover are compared on a consistent basis. The assumption is that each of pretreatment technologies has same amount lignin residues with production cost. The process model of PHA fermentation was embedded in NREL base model so that systematic effects of lignin reactivity on PHA fermentation were accounted in the overall process. Economic drivers influenced by lignin valorization process are yield of PHA, chemical costs, and lignin treatment costs[76]. An Aspen Plus process flow diagram describing the production of PHAs from lignin streams is shown in Supplementary Figs. 12–16 and Supplementary Tables 3–7.

**Analysis methods.** Composition analysis including carbohydrates and lignin was conducted according to the Laboratory Analysis Procedures (LAP) of the NREL. Sugars in the solid and liquid fraction were analyzed by an Ultimate 3000 HPLC System (Thermo Scientific, USA) equipped with an Aminex HPX-87P carbohydrate analysis column (Bio-Rad Laboratories, CA) and a refractive index detector using HPLC grade water as the mobile phase at a flow rate of 0.6 ml/min. PHA yield in fermentation was calculated based on the consumption of substrate in medium including both lignin and residual glucose by *P. putida* KT2440. Mass balance was carried out in the whole fractionation process. Sugar yield in the whole process was calculated based on the corn stover feedstock used for each pretreatment. Error bars in the figures represented the standard deviation of the replicates.

## Data availability
The experiment data that support the findings of this study are available as a supplement to this manuscript (Supplementary figures and tables). Any other relevant data are available from the authors upon reasonable request. Source data are provided with this paper.

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

## Acknowledgements

The work was financially supported by the U.S. DOE (Department of Energy) EERE (Energy Efficiency and Renewable Energy) BETO (Bioenergy Technology Office) (grant no. DE-EE0006112, DE-EE0007104, and DE-EE0008250).

## Author contributions

Z.H.L. and J.S.Y. wrote and revised the manuscript. Z.H.L performed the experiment and data analyses, N.J.H., Y.Y.W., and A.J.R. performed the lignin characterization, C.D. and R.B. prepared the steam explosion samples, F.R.L. participated in the fermentation experiment, R.C.S. and B.Y. performed the techno-economic analysis, D.B.H. prepared the liquid hot water samples, B.E.D. prepared the AFEX samples. All authors reviewed and approved the final manuscript.

## Competing interests

The authors declare no competing interests.
