## [Peer Review File · Nature Communications]

REVIEWER COMMENTS

Reviewer #1 (Remarks to the Author):

The manuscript investigated the feasibility of adding NaOH treatment (as a second pretreatment) after pretreatment. The manuscript contains interesting results. However, it is premature to be published in a current form.

Name of the process - PIP. It is not clear why it needs to be called “plug-in processes”. In essence, it is a second pretreatment. Authors might want to have an attractive name (PIP), but its justification is weak. Unless there is a strong justification, it would be simpler to say “alkaline treatment as second process” or something like that.

Results are not adequately presented. Since the manuscript describes the overall process (biomass to final product), there should be a table summarizing mass balance at each step for five scenarios. Starting from 100 grams (or 100 metric tons) of biomass, how many grams of glucan coming into enzymatic hydrolysis, and how many grams of glucose is obtained. Without this overall mass balance, the advantage of the proposed process is challenging to understand.

Conversion yield not significant. There are several sentences throughout the manuscript, saying “xx content after PIP increased/decreased by a%, b%, c%, and d%”. Cellulose conversion (or lignin conversion) does not provide an overall picture of the process. Again, the actual mass balance at each stage is necessary to understand the proposed process.

NaOH amount at second treatment. Based on the description on Page 7, 50 g of solid (dw) is used at “10% solid with 1% NaOH”. Therefore, it seems the amount of NaOH is 4.5 g. So, the amount of NaOH is not insignificant. Is there a NaOH recovery process? What is the pH after NaOH mixing after pretreatments? Solid from DSA and LHW are very acidic, so they might need more NaOH to make the same pH.

Temperature at second treatment. Temperature (121C) seems too high, and some carbohydrates will be lost due to a peeling reaction. Any reason to choose this temperature? How much was the peeling reaction observed?

PHA product recovery. The details in PHA purification and drying are not specified in the manuscript. How much product recovery was experimentally performed after fermentation? What process was included in Aspen modeling work?

Techno-economic analysis. Details in economic analysis are mostly missing in the manuscript. For example, authors can add a separate table for operation cost, including the amount of NaOH purchase, wastewater treatment, PHA purification and others. The detailed information is necessary to compare five different scenarios, instead of presenting one value (MSP).

Reviewer #2 (Remarks to the Author):

This is a difficult paper to review at least for this journal. I can find little to fault with the paper - it contains a well argued and strongly supported case for altering the unit process design of bio-refineries. The data is clear and persuasive. The article is well written and generously illustrated. I have no doubt of the value to and potential impact for the designers of bio-refineries but are these likely to read Nature communications? There is also a question over appeal since while the bio-economy is attracting more and more attention and hence more and more research, it is probably still only a narrow section of the science community that is seriously interested in bio-refineries especially from the point of view of unit operations and pre-treatment options of biomass. Overall I think this belongs in a more specialist/technical journal although the editor may think differently.

Reviewer #3 (Remarks to the Author):

In this manuscript, new processes were combined and designed as PIP for improving the economy of Biorefinery. However, there are some problems that need to be addressed.

1, plug-in processes seems to be a technology integration, but this concept were not well explained and defined in the manuscript. Please explain the related in the introduction part in detail.

2, In figure 5, It seems like the microbe used in this research mainly utilize glucose but not lignin. How to produce ethanol when there were no glucose left?

3, In figure 6, what exactly is "substrate", is that is glucose or lignin?

Response to Reviewer #1 (Remarks to the Author):

Answers to general comments of reviewer #1:

The manuscript investigated the feasibility of adding NaOH treatment (as a second pretreatment) after pretreatment. The manuscript contains interesting results. However, it is premature to be published in a current form.

Answer: we would like to thank the reviewer for thorough reading of this manuscript. We also really appreciate that the reviewer highlighted the results of the manuscript and regard the results as ‘interesting’. We have carried out both experiments and major revision to address reviewer’s comments. In particular, we have rewritten and refined the manuscript to highlight the scientific significance and implications of the findings. The reviewer’s constructive suggestions have significantly improved the quality of the manuscript as described below. The revisions have been given in the revised manuscript correspondingly.

Answers to specific comments of Reviewer #1:

1: to comment 1: Name of the process - PIP. It is not clear why it needs to be called “plug-in processes”. In essence, it is a second pretreatment. Authors might want to have an attractive name (PIP), but its justification is weak. Unless there is a strong justification, it would be simpler to say “alkaline treatment as second process” or something like that.

Answer: we appreciate the comment to guide us to improve clarity. Based on the reviewer’s suggestion, we have changed the name of our process to be ‘Plug-in Processes of Lignin (PIPOL)’ to be more precise. We understand the reviewer’s confusion probably arises from traditional pretreatment perspective, considering that the ‘Plug-in Processing of Lignin (PIPOL)’ thus improves the carbohydrate process efficiency, like any pretreatment technology. However, PIPOL is not just a ‘second pretreatment’ or ‘alkaline treatment as second process’. PIPOL is fundamentally different from pretreatment in terms of its definition, purpose, biorefinery integration, delivery and outcome. We hereby discuss these aspects as follows.

First, the ‘Plug-in processes of lignin (PIPOL)’ refers to the processes including

solubilization, conditioning and fermentation of lignin, so that lignin can be efficiently processed for the biological lignin valorization to PHAs. It is not a single step ‘treatment’ or ‘pretreatment’, but the entire multiple-step processes. The rationale for the term ‘plug-in’ lies in the fact that the processes were integrated with current biorefinery to establish new scenarios. Considering the integration with biorefinery and the involvement of lignin bioconversion, we therefore adopt the reviewer’s suggestion to rename the process more precisely as ‘plug-in processes of lignin (PIPOL)’.

Second, the purpose of PIPOL concept is to achieve efficient processing of both carbohydrate and lignin, addressing the dilemma of ‘carbohydrate-first’ and ‘lignin-first’ biorefinery configuration. The synergistic improvement of carbohydrate and lignin conversion efficiency is different from either traditional ‘carbohydrate-first’ pretreatments or the recently proposed ‘lignin-first’ processes. Most traditional pretreatments have been built for deconstruction of biomass to render more processible carbohydrates, and the lignin is left as an underutilized biorefinery waste. Recently, biological lignin valorization has been shown with significant potential to convert heterogeneous aromatics via ‘biological funneling’ to produce lignin-based products (Supplementary Fig. 8) [Ragauskas, A.J. et al. *Lignin Valorization: Improving Lignin Processing in the Biorefinery*. *Science* 344, 709 (2014); Fuchs, G., Boll, M. & Heider, J. *Microbial degradation of aromatic compounds - from one strategy to four*. *Nat Rev Microbiol* 9, 803-816 (2011); Linger, J.G. et al. *Lignin valorization through integrated biological funneling and chemical catalysis*. *Natl Acad Sci USA* 111, 12013-12018 (2014); Liu, Z.H. et al. *Identifying and creating pathways to improve biological lignin valorization*. *Renew Sust Energ Rev* 105, 349-362 (2019)]. The biological lignin valorization to valuable products such as PHAs could transform biorefinery with better economics and sustainability and address the environmental challenges of non-degradable plastics. Even though classic pretreatments could impact the structure and activity of the lignin, they are not optimized for lignin processing. PIPOL addresses this challenge to integrate a lignin bioproduct stream in current biorefinery. On the other hand, PIPOL is different from the recent concept of lignin-first biorefinery, where the biorefinery design will optimize to preserve the β -O-4 linkages and achieve less condensed structure for downstream processing [Abu-Omar;

M.M. et al. Guidelines for performing lignin-first biorefining. Energ Environ Sci (2021); Renders, T., Van den Bosch, S., Koelewijn, S.F., Schutyser, W. & Sels, B.F. Lignin-first biomass fractionation: the advent of active stabilisation strategies. Energ Environ Sci 10, 1551-1557 (2017); Questell-Santiago, Y.M., Galkin, M.V., Barta, K. & Luterbacher, J.S. Stabilization strategies in biomass depolymerization using chemical functionalization. Nat Rev Chem 4, 311-330 (2020).]. Lignin-first fractionation produces high molecular weight molecules not suitable for bioconversion. PIPOL thus is a set of processes to achieve lignin fractionation and bioconversion to achieve synergistic lignin and carbohydrate conversion at high efficiency, which is beyond the concept of pretreatment alone.

Third, the configuration of PIPOL is different based on various pretreatments. PIPOL can be integrated with the pretreatment technologies with different solid and liquid handling based on the biorefinery designs to enable efficient lignin utilization. In order to highlight the differences, we have modified the Fig. 1 as the ‘Plug-in processes of lignin (PIPOL)’ to clearly define how different solubilization, conditioning, and fermentation steps were incorporated into the new biorefinery scenarios (Fig. 1 b-f) as compared with current biorefinery configuration (Fig. 1a). We further carried out systemic performance analysis of these new biorefinery scenarios.

Fourth, the deliverable is not just a secondary pretreatment, but an effective process for biological lignin valorization, which can be incorporated into current biorefinery to improve the economics and sustainability. As expected, the new biorefinery scenarios by incorporating lignin valorization through PIPOL synergistically improved the biological processing of lignin by enhancing its solubility and reactivity, enhanced the accessibility of carbohydrate, maximized the overall output of products, and minimized the costs of lignin upgrading. Therefore, the new PIPOL-integrated biorefinery designs enabled more complete biomass utilization, improved overall carbon conversion efficiency, and enabled better biorefinery sustainability.

Considering all these aspects and the reviewer’s suggestion, we have changed the name of the process as ‘**Plug-in processes of lignin (PIPOL)**’ to be more specific. We also revised Fig. 1 and the manuscript substantially to clarify the concept and avoid any confusion as a second pretreatment step.

Fig. 1 Flow process diagram of the biorefinery scenarios employed leading pretreatments with the incorporation of ‘plug-in processes of lignin (PIPOL)’ (purple module). PIPOL integrates the dissolution, conditioning, and bioconversion of lignin as indicated in different biorefinery scenario as follows. a, a general biorefinery process for ethanol production; b, PIPOL-integrated biorefinery scenario with dilute sulfuric acid pretreatment (DSA); c, PIPOL-integrated biorefinery scenario with steam explosion pretreatment (SEP); d, PIPOL-integrated biorefinery scenario with liquid hot water pretreatment (LHW); e, PIPOL-integrated biorefinery scenario with ammonia fiber expansion (AFEX); f, PIPOL-integrated biorefinery scenario with sodium hydroxide pretreatment (SHP). L in orange square represents lignin, H in light green square represents hemicellulose, C in blue square represents cellulose, P in green circle represents product, S-L represents solid-liquid separation. (Page 34)

The present study aims to address these challenges with the design of a novel ‘Plug-In Processes of lignin (PIPOL)’ for biological lignin valorization that can be directly incorporated into current biorefinery. The research systemically evaluated lignin bioconversion performance on the integration of PIPOL with five leading pretreatments in biorefinery. The PIPOL had been designed

with the integration of solubilization, conditioning and fermentation to improve the solubility and reactivity of lignin and its microbial conversion in biorefinery designs. **(Pages 5 and 6)**

The fundamental challenges for the bioconversion of biorefinery waste lies in the fact that most of the current pretreatment and hydrolysis platforms generate a largely solid lignin waste stream. Our previous studies have established that lignin dissolution is critical for bioconversion, as aromatic monomers and oligomers can serve as substrates for microbes, but not the larger molecules. **Based on this understanding, we have designed a set of procedures, dubbed as ‘plug-in processes of lignin (PIPOL)’, to integrate the solubilization, conditioning, and fermentation for lignin bioconversion. PIPOL is incorporated with current biorefinery to achieve lignin dissolution and bioconversion (Fig. 1a-f).** Considering the features of different leading pretreatments, the PIPOL designs were implemented with various biorefinery configurations and compared in terms of lignin conversion, PHA yield, and economics. **(Pages 6 and 7)**

In detail, five leading pretreatments that have the potential to be commercially implicated were employed to maximally unleash their roles in the improvements of deconstruction efficiency, lignin processibility, and biorefinery performance. As the soluble low-molecular-weight lignin will enable its microbial conversion, the solubilization of the pretreated solid was developed to tailor lignin chemistry and enhance its biological processibility. The soluble lignin carried specific reactivates and can be converted to PHAs by fermentation after conditioning. **The processing steps of solubilization, conditioning, and fermentation for lignin bioconversion were designed as ‘PIPOL’ to be incorporated into a biorefinery.** For acidic pretreatments of DSA and LHW (Fig. 1b and 1d), the lignin solubilization was conducted after separating the pretreated solid in the pretreatment. The soluble lignin stream from the solubilization can be mixed well with the liquid stream from pretreatment to eliminate the conditioning of lignin stream. This PIPOL design integrated the solubilization and fermentation for lignin bioconversion in a biorefinery, where carbohydrates were enzymatically hydrolyzed after the lignin solubilization. As SEP and AFEX were carried out at high solids, the PIPOL design integrated the solubilization, conditioning, and fermentation following the pretreatment (Fig. 1c and 1e). For the alkaline approach, SHP was employed to directly fractionate the lignin from LCB for microbial conversion (Fig. 1f). **(Page 7)**

The operations of solubilization, conditioning and fermentation were designed to improve lignin solubility and reactivity for microbial conversion (Fig. 1). As these three steps were directly

incorporated into current biorefinery to achieve co-utilization of lignin and carbohydrate at high efficiency. They were named as ‘Plug-In Processes of Lignin (PIPOL)’. (Page 20)

2: to comment 2: Results are not adequately presented. Since the manuscript describes the overall process (biomass to final product), there should be a table summarizing mass balance at each step for five scenarios. Starting from 100 grams (or 100 metric tons) of biomass, how many grams of glucan coming into enzymatic hydrolysis, and how many grams of glucose is obtained. Without this overall mass balance, the advantage of the proposed process is challenging to understand.

Answer: we really appreciate the comment and agreed with reviewer. As suggested by the reviewer, we have provided the overall mass balance to highlight the advantage of the PIPOL according to reviewer’s suggestions. The mass balance of biomass transformation had been given in **Supplementary Fig. 7 (Pages 49-50)**. We have also provided more discussion in the revised manuscript as follows.

Supplementary Fig. 7 Mass balance for the fractionation processes of corn stover biomass by integrating leading pretreatments with the ‘plug-in processes of lignin (PIPOL)’. a, PIPOL-integrated biorefinery scenario with dilute sulfuric acid pretreatment (DSA); b, PIPOL-integrated biorefinery scenario with steam explosion pretreatment (SEP); c, PIPOL-integrated biorefinery scenario with liquid hot water pretreatment (LHW); d, PIPOL-integrated biorefinery scenario with ammonia fiber expansion (AFEX); e, PIPOL-integrated biorefinery scenario with sodium hydroxide pretreatment (SHP). (Pages 49-50)

Supplementary Fig. 3 The leading pretreatment and the ‘plug-in processes of lignin (PIPOL)’ improved sugar yields from corn stover biomass. DSA, dilute sulfuric acid pretreatment; SEP, steam explosion pretreatment; LHW, liquid hot water pretreatment; AFEX, ammonia fiber expansion; SHP, sodium hydroxide pretreatment. The conditions of pretreatment were provided in Table 1. (Page 45)

3: to comment 3: Conversion yield not significant. There are several sentences throughout the manuscript, saying “xx content after PIP increased/decreased by a%, b%, c%, and d%”. Cellulose conversion (or lignin conversion) does not provide an overall picture of the process. Again, the actual mass balance at each stage is necessary to understand the proposed process.

Answer: we really appreciated the comment and made the substantial changes to the manuscript according to the suggestions. In answering to previous comment, we have provided the detailed mass balance that includes lignin transformation in the solid and liquid stream during pretreatment and solubilization. Lignin yield in pretreatment and solubilization processes had also been presented in Fig. 2 (Page 35). To highlight and compare the effects of solubilization on both carbohydrate and lignin transformation in biomass, we monitored the component content in solid fraction from pretreatment and solubilization process in liquid fraction in Supplementary Fig. 1 (Page 43). The glucan and xylan conversion had been given in Fig. 3 (Page 36).

As suggested by reviewer, we have provided the mass balance data of the entire processes of PIPOL-integrated biorefinery in Supplementary Fig. 7 (Pages 49-50). This

mass balance could provide an overall picture of the process and clearly showed the cellulose and lignin transformation. More discussion on glucan conversion and lignin transformation was also provided in the revised manuscript according to reviewer's suggestions.

4: to comment 4: NaOH amount at second treatment. Based on the description on Page 7, 50 g of solid (dw) is used at "10% solid with 1% NaOH". Therefore, it seems the amount of NaOH is 4.5 g. So, the amount of NaOH is not insignificant. Is there a NaOH recovery process? What is the pH after NaOH mixing after pretreatments? Solid from DSA and LHW are very acidic, so they might need more NaOH to make the same pH.

Answer: we appreciate the comment very much. The solubilization of pretreated solids were carried out at 10% (w/w) solid loading with 1% NaOH (w/w) under 121 °C for 30 min (Page 20). The reviewer is largely right on the NaOH consumption. The consumption of NaOH used in PIPOL have been calculated in the techno-economic analysis (TEA) and more information had been provided in the supplemental materials (Pages 58 and 62).

We agree with the reviewer's suggestion on NaOH recovery very much. In this study, we evaluated the efficient dissolution and conversion of lignin. The NaOH treatment step of PIPOL was designed to deconstruct lignin polymer to generate lignin fraction with more hydrophilic groups, smaller molecules, and less condensed structures, which subsequently improved the solubility, reactivity, and processibility of lignin and facilitated its bioconversion to PHAs. The current TEA took into consideration of NaOH usage and still showed a competitive price. Nevertheless, we agreed with the reviewer that NaOH recovery process could reduce the cost of PHA production, though it is not the focus of the present study. We will follow the reviewer's suggestion to develop the recovery process of NaOH in the future study and evaluate how the recovery process of NaOH can contribute to the reduction of capital cost with the TEA. Such study will take extra effort of process design, implementation, and technoeconomic analysis, which is beyond the scope of this publication.

As suggested by reviewer, we monitored the pH value of the liquid streams

(Supplementary Fig. 4, Page 46). After the pretreatment and solubilization step, the pH value of the lignin liquid stream varied with the pretreatment employed. Results suggested that the pH value of soluble lignin stream depended on the pretreatment used for corn stover biomass. As shown in Supplementary Fig. 4, the pH value of soluble lignin stream for PIPOL was around 11.9, 11.5, 10.8, 12.8, and 9.2 after the solubilization of the pretreated solid prepared from DSA, SEP, LHW, AFEX, SHP, respectively.

The reviewer's question regarding to the impact of pretreatment technologies on the pH after NaOH mixing is also a very in-depth one. As shown in Table 1, the low severity was used for DSA and LHW pretreatments in this study. In fact, the severity selection already took into the consideration of the integration with PIPOL when we first design the multi-stream integrated biorefinery. The dilute sulfuric acid pretreatment was conducted under 150 °C for 10 min with 0.2% H₂SO₄, while liquid hot water carried out under 160 °C for 5 min. These pretreatment severities are lower than those employed in previous studies. The conditions of dilute sulfuric acid pretreatment usually used in previous studies were 140-180 °C for 20-40 min with 1% H₂SO₄, while the conditions of liquid hot water employed in previous studies were 180-200 °C for 10-60 min.

In fact, a relatively weaker pretreatment in combination with the PIPOL will still achieve very effective carbohydrate hydrolysis, yet also allowing a liquid lignin stream for fermentation. PIPOL was designed to integrate with the pretreatments to synergistically deconstruct the biomass in the present study. When designing the biorefinery integration, we actually took into the consideration of both pretreatment and PIPOL, as well as both lignin and carbohydrate stream. The goal of our holistic biorefinery design is that the integration of PIPOL and the pretreatment with low severity can synergistically improve the biological processing of lignin and the enzymatic accessibility of carbohydrate. A too severe pretreatment could generate more condensed lignin and inhibitors to prevent efficient carbohydrate and lignin fermentation. On the other hand, the alkaline dissolution step of PIPOL could also help to reduce biomass recalcitrance, alleviating the need for high severity in pretreatment.

With all these considerations, we have used relatively low severity in DSA and LHW. The result highlighted that lower pretreatment severity did not impact the effective utilization of carbohydrates, when combined with PIPOL. A side benefit is that the pH for the dissolute lignin is not too extreme, making it more ready for conditioning.

The pH value of the liquid stream from acidic pretreatment depends on acid, holding temperature and residence time employed. Because of the low pretreatment severity of DSA and LHW employed in this study, the pH value of liquid stream is only 2.7 and 3.3, respectively. Such pH is much higher than that in the traditional pretreatment conditions, which could be less than 1.0 in previous studies. On the one hand, the solid fraction from DSA and LHW can be separated from the liquid stream, after DSA and LHW pretreatment for the solubilization of lignin (Fig. 1). The solid fraction was dissolved into water with the solid loading of 10% and then used for further solubilization of lignin. As the solid fraction contains very little acidic liquids from pretreatment, it will consume limited amount of NaOH to neutralize the mixture and to achieve effective dissolution of lignin from this fraction. More importantly, the alkaline lignin solution can be further used to mix with acidic liquid stream for neutralization, and no extra alkaline is needed. Taken together, the advantages of this process for DSA and LHW with relatively weak severity and PIPOL integration are: 1) achieving highly efficient carbohydrate hydrolysis; 2) deriving a soluble lignin fraction with high bioprocessibility; 3) synergizing the lignin dissolution from solid fraction with neutralization of acidic liquid stream to reduce NaOH usage and to achieve process intensification; 4) reducing the inhibitors for hydrolysis, fermentation, and lignin bioconversion; and 5) potentially reducing condensed lignin and pseudo-lignin.

We really appreciate reviewer's comments to guide us to improve clarity. The revision has also been provided in the revised manuscript correspondingly to improve clarity, including the updated Fig. 1 to highlight the solid and liquid handling and the pH value of different streams.

Reference:

Wyman CE, Balan V, Dale BE, Elander RT, Falls M, Hames B, Holtzapple MT, Ladisch MR, Lee YY, Mosier N, Pallapolu VR. Comparative data on effects of leading pretreatments and enzyme

loadings and formulations on sugar yields from different switchgrass sources. *Bioresource technology*. 2011 Dec 1;102(24):11052-62.

Kumar R, Mago G, Balan V, Wyman CE. Physical and chemical characterizations of corn stover and poplar solids resulting from leading pretreatment technologies. *Bioresource technology*. 2009 Sep 1;100(17):3948-62.

Liu ZH, Qin L, Li BZ, Yuan YJ. Physical and chemical characterizations of corn stover from leading pretreatment methods and effects on enzymatic hydrolysis. *ACS Sustainable Chemistry & Engineering*. 2015 Jan 5;3(1):140-6.

Supplementary Fig. 4 The pH value of the liquid stream of leading pretreatment and the soluble lignin stream fractionated from the biorefinery scenario integrated 'plug-in processes of lignin (PIPOL)' of biological lignin valorization. DSA, dilute sulfuric acid pretreatment; SEP, steam explosion pretreatment; LHW, liquid hot water pretreatment; AFEX, ammonia fiber expansion; SHP, sodium hydroxide pretreatment. **(Page 46)**

Fig. 1 Flow process diagram of the biorefinery scenarios employed leading pretreatments with the incorporation of ‘plug-in processes of lignin (PIPOL)’ (purple module). PIPOL integrates the dissolution, conditioning, and bioconversion of lignin as indicated in different biorefinery scenario as follows. a, a general biorefinery process for ethanol production; b, PIPOL-integrated biorefinery scenario with dilute sulfuric acid pretreatment (DSA); c, PIPOL-integrated biorefinery scenario with steam explosion pretreatment (SEP); d, PIPOL-integrated biorefinery scenario with liquid hot water pretreatment (LHW); e, PIPOL-integrated biorefinery scenario with ammonia fiber expansion (AFEX); f, PIPOL-integrated biorefinery scenario with sodium hydroxide pretreatment (SHP) (Page 34)

Here, we use a ligninolytic organism, *P. putida* KT2440, to demonstrate that the soluble lignin from the PIPOL can be used to synthesize the PHAs. Microbial conversion performance can be promoted with the improved processibility of lignin (Fig. 4). The soluble lignin stream has a high pH value, and the exact pH depended on the pretreatment employed (Supplementary Fig. 4). Generally, a neutralization step was needed to adjust pH to facilitate the bacteria growth. After conditioning to an optimal pH ~7.0 for *P. putida* KT2440, the soluble lignin stream was evaluated for PHA fermentation. (Page 10)

5: to comment 5: Temperature at second treatment. Temperature (121C) seems too high, and some carbohydrates will be lost due to a peeling reaction. Any reason to choose this temperature? How much was the peeling reaction observed?

Answer: we appreciate the comment very much. The mild conditions (121 °C) of the lignin solubilization were carefully chosen based on literature review and our previous experience, so that it is optimal for both lignin fractionization and downstream bioconversion. We agreed with the reviewer that several factors need to be taken into considerations. These factors includes avoiding significant sugar degradation, minimizing condensed lignin, effective lignin dissolution, and reducing inhibitors for downstream conversion. Taking into these considerations, the solubilization was conducted with mild conditions of 121 °C. We have provided the mass balance data in the solubilization for each biorefinery (Supplementary Fig. 7, Pages 49-50). Results showed that there is no serious degradation of carbohydrates during lignin solubilization at 121 °C. To provide more thorough response to the reviewer, we hereby further expand our rationale in the design of this temperature.

First, the temperature (121 °C) of the solubilization process is lower than that of the traditional pretreatment (160~220 °C). Generally, the carbohydrates from lignocellulosic biomass would not be degraded at 121 °C in pretreatment. As shown in the mass balance, the mild temperature condition at 121°C promoted the lignin dissolution, yet avoided the serious degradation of glucan and xylan existed in the pretreated solid fraction. Such temperature did not generate much byproducts that could become inhibitors for downstream fermentation, too.

Second, the pretreatment integrated with the solubilization at low holding temperature can improve the yields of fermentable sugars. The carbohydrate in biomass is heterogeneous in their processibility. Some parts are more readily to be hydrolyzed, whereas others are more difficult. The pretreatment with a holding temperature of 140-200 °C can remove these readily hydrolyzed carbohydrates, thus the resistance carbohydrate could not be seriously degraded in the solubilization with a holding temperature of 121 °C. We have provided the mass balance data of the lignin solubilization in Supplementary Fig. 7 and the results showed that few cellulose and hemicellulose were degraded in the lignin solubilization process when the holding temperature of 121 °C were employed.

References

Jacobsen SE, Wyman CE. Cellulose and hemicellulose hydrolysis models for application to current and novel pretreatment processes. In *Twenty-first symposium on biotechnology for fuels and chemicals 2000* (pp. 81-96). Humana Press, Totowa, NJ.

Shi S, Guan W, Kang L, Lee YY. Reaction kinetic model of dilute acid-catalyzed hemicellulose hydrolysis of corn stover under high-solid conditions. *Industrial & Engineering Chemistry Research*. 2017 Oct 4;56(39):10990-7.

Negahdar L, Delidovich I, Palkovits R. Aqueous-phase hydrolysis of cellulose and hemicelluloses over molecular acidic catalysts: Insights into the kinetics and reaction mechanism. *Applied catalysis B: environmental*. 2016 May 5;184:285-98.

Third, the solubilization at 121 °C have improved the solubility, reactivity and processibility of lignin for biological conversion. The soluble lignin yield reached to 70.2%, 65.6%, 56.2%, 81.5% and 48.7% from DSA-, SEP-, LHW-, AFEX- and SHP-treated solids, respectively. After pretreatment, the solubilization process depolymerized lignin to low molecular weight derivatives with less β -O-4 and β - β linkages, more phenolic hydroxyl and carboxyl groups, which resulted in the improved bioreactivity and processibility by ligninolytic *P. putida* KT2440 strain. Most importantly, the holding temperature used in the solubilization process reduced the repolymerization of lignin and thus avoid the generation of condensed lignin, which is harmful to bioconversion performance as condensed lignin cannot be readily consumed by ligninolytic *P. putida* KT2440 strain.

Overall, by integrating with pretreatment, the solubilization under mild conditions (121°C) allowed the maximized depolymerization and dissolving of lignin, avoided the generation of condensed lignin and by-products as inhibitors, and synergistically increased carbohydrate output. The corresponding revision has also been provided in this revised manuscript.

For lignin bioconversion, the solubilization of the pretreated solid was conducted to produce soluble lignin and to improve lignin reactivity for microbial conversion. In detail, 50 g pretreated solid (dw) was loaded into a 1.0-L screw neck bottle with a polypropylene pour ring at 10% (w/w) solid loading with 1% NaOH (w/w). To avoid the significant degradation of sugars and minimize the generation of condensed lignin, the solubilization was conducted with mild conditions of 121 °C

for 30 min using Amsco LG 250 Laboratory Steam Sterilizer (Steris, USA). The heating time was about 5 min, while the cooling time was about 10 min. The slurry was vacuum filtrated through a Brinell funnel to separate the soluble lignin stream from the solid residues. After conditioning, the liquid stream containing soluble lignin was used as carbon source in fermentation. The operations of solubilization, conditioning and fermentation were designed to improve lignin solubility and reactivity for microbial conversion (Fig. 1). As these three steps were directly incorporated into current biorefinery to achieve co-utilization of lignin and carbohydrate at high efficiency. They were named as ‘Plug-In Processes of Lignin (PIPOL)’. (Page 20)

6: to comment 6: PHA product recovery. The details in PHA purification and drying are not specified in the manuscript. How much product recovery was experimentally performed after fermentation? What process was included in Aspen modeling work?

Answer: we appreciate the comment very much. In the present study, we employed the method widely used in PHAs analysis in literatures to determine the PHA content based on dried cell biomass of *P. putida* strains.

As for PHA extraction, TEA in this ASPEN model employed the very standard industrial process of PHA extraction and recovery. We have provided the detailed information of TEA with the integration of PHA recovery in the revised manuscript. The PHA quantification has an inherent extraction step similar to industrial process and ASPEN model. The purification and recovery of PHAs has already been taken into the consideration in the TEA mode. PHA extraction is a mature industrial process with known high yield. The detailed information of TEA for PHAs extraction had been shown as follows.

Supplementary Fig. 14 Aspen model of the flow diagram of the PHA extraction operation. This process is responsible for the recovery of the PHAs produced intracellularly in *P. putida* KT2440. A 12-hr batch cell powders processed from fermentation is being processed. Extraction of the PHAs is commenced by cell disruption of the cell powders by introducing ethyl acetate in a continuously stirred batch tank. Two cell precipitation tanks were needed to accommodate disruption of the 12-hr batch cell powders. After cell disruption, aqueous solution is introduced into the filter press and centrifuge for subsequent removal of insoluble solids. Ethyl acetate containing PHA flocculates is then subjected for precipitation. In this operation, 3:1 hexane to ethyl acetate volume ratio is required. After precipitation, PHA polymers are separated using a 2 µm filter press. With this volume of the required solvent in the extraction process, a solvent recovery is definitely a must. After PHA polymers are dewatered it will be washed to remove residual salts. Right after washing, PHAs will be dried in a vacuum dryer. Dried PHA is now ready for storage and/or marketing. (**Page 57**)

7: to comment 7: Techno-economic analysis. Details in economic analysis are mostly missing in the manuscript. For example, authors can add a separate table for operation cost, including the amount of NaOH purchase, wastewater treatment, PHA purification and others. The detailed information is necessary to compare five different scenarios, instead of presenting one value (MSP).

Answer: we really appreciate the comment. As suggested by the reviewer, more detailed information of techno-economic analysis (TEA) have been provided in **Supplementary information (Supplementary Fig. 10-14 and Supplementary Tables 1-5, Pages 53-62)**. We have also added a separate table for financial assumptions and design basis including operation cost and others (**Table 3, Page 33; Supplementary Table 1, Page 58**). The corresponding revisions have also been provided in the revised manuscript as follows.

Supplementary Fig. 10 Aspen Plus process flow diagram of the ‘plug-in processes of lignin (PIPOL)’ of biological lignin valorization to produce polyhydroxyalkanoates (PHAs), which includes lignin treatment, PHA fermentation, cell recovery, and PHA extraction. (**Pages 53-57**)

Table 3 Techno-economic analysis (TEA) of the biological lignin valorization to polyhydroxyalkanoates (PHAs) designed as ‘plug-in processes of lignin’ in a biorefinery (**Page 33**)

	Scenario 1 DSA- PIPOL	Scenario 2 SEP- PIPOL	Scenario 3 LHW- PIPOL	Scenario 4 AFEX- PIPOL	Scenario 5 SHP- PIPOL
Annual production/MMkg	2.19	3.21	2.6	3.54	2.02
Total capital cost/MM\$	44.98	42.64	42.13	39.85	42.12
Total operation cost/MM\$/yr	11.2	12.42	12.26	13.24	14.65
Raw material/MM\$/yr	4.88	5.9	5.77	7.4	8.12
Utilities/MM\$/yr	0.268	0.267	0.266	0.27	0.266
Unit cost/\$/kg	7.52	5.49	6.73	5.05	9.9
Rate of return/%	10	10	10	10	10
Minimum selling price/\$/kg	9.58	6.82	8.32	6.18	11.99

Supplementary Table 1 Financial assumptions and design basis (**Page 58**)

Case	Value
Plant life	30 years
Cost year (dollar unit)	2014 dollars
Capacity factor	90%

Discount rate	10%
General plant depreciation	200% declining balance (DB)
General plant recovery period	7 years
Steam plant depreciation	150% DB
Steam plant recovery period	20 years
Federal tax rate	35%
Financing	40% equity
Loan terms	10 year loan at 8% APR
Construction period	3 years
First 12 months' expenditures	8%
Next 12 months' expenditures	60%
Last 12 months' expenditures	32%
Working capital	5% of fixed capital investment
Start-up time	3 months
Revenues during start-up	50%
Variable costs during start-up	75%
Fixed costs during start-up	100%
Lignin (\$/ton)	55.8 ^[1]
H ₂ SO ₄ (\$/ton)	81.39 ^[2]
NaOH (\$/ton)	135.65 ^[2]
Ethyl Acetate (\$/ton)	1100 ^[2,3]
Hexane (\$/lb)	0.5394 ^[2,3]
Ethanol (\$/gal)	2.051 ^[2,3]

Supplementary Table 2 Mass flow of key streams in the conditioning and fermentation sections

(Pages 59-62)

Component	Units	101	109	110	114	123
Total flow	kg/hr	22375.437	71664.840	83736.240	83737.440	83737.440
Water	kg/hr	7836.510	57122.310	57122.310	57122.740	57126.480
Ethanol	kg/hr	3.493	3.493	3.493	3.493	3.493
Glucose (SS)	kg/hr	17.822	17.822	17.822	16.040	4.812
Galactose (SS)	kg/hr	26.082	26.082	26.082	26.082	26.082
Mannose (SS)	kg/hr	10.944	10.944	10.944	10.944	10.944
Xylose (SS)	kg/hr	26.527	26.527	26.527	26.527	26.527
Arabinose (SS)	kg/hr	5.016	5.016	5.016	5.016	5.016
Glucooligomers (SS)	kg/hr	31.788	31.788	31.788	31.788	31.788
Galactooligomers (SS)	kg/hr	0.652	0.652	0.652	0.652	0.652
Mannooligomers (SS)	kg/hr	0.274	0.274	0.274	0.274	0.274
Extractives (SS)	kg/hr	278.064	278.064	278.064	278.064	278.064
Lignisol (SS)	kg/hr	14.965	14.965	14.965	14.965	14.965
HMF	kg/hr	8.404	8.404	8.404	8.404	8.404
Furfurals	kg/hr	3.331	3.331	3.331	3.331	3.331

Lactic acid	kg/hr	38.607	38.607	38.607	38.607	38.607
Xylitol	kg/hr	18.553	18.553	18.553	18.553	18.553
Glycerol	kg/hr	0.528	0.528	0.528	0.528	0.528
Succinic acid	kg/hr	1.848	1.848	1.848	1.848	1.848
Ammonia	kg/hr	-	3.600	3.600	3.600	3.600
Ammonium sulfate	kg/hr	56.560	56.560	56.560	56.560	56.560
Ammonium acetate	kg/hr	42.422	42.422	42.422	42.422	42.422
DAP	kg/hr	2.970	2.970	2.970	74.346	74.346
O ₂	kg/hr	0.328	0.328	2795.332	2795.332	168.410
N ₂	kg/hr	0.585	0.585	9205.581	9205.581	9205.581
CO ₂	kg/hr	-	-	-	-	4464.517
PHA (IS)	kg/hr	-	-	-	-	2263.076
P. Putida	kg/hr	-	-	-	2.662	2.662
Cell mass (IS)	kg/hr	-	-	-	-	1775.836
Cellulose (IS)	kg/hr	768.559	768.559	768.559	768.559	768.559
Galactan (IS)	kg/hr	18.218	18.218	18.218	18.218	18.218
Mannan (IS)	kg/hr	7.644	7.644	7.644	7.644	7.644
Xylan (IS)	kg/hr	478.485	478.485	478.485	478.485	478.485
Arabinan (IS)	kg/hr	58.310	58.310	58.310	58.310	58.310
Lignin (IS)	kg/hr	7336.278	7336.278	7336.278	7336.278	1467.256
Protein (IS)	kg/hr	1684.852	1684.852	1684.852	1684.769	1684.769
Ash	kg/hr	2415.700	2415.700	2415.700	2415.700	2415.700
Enzyme (IS)	kg/hr	306.269	306.269	306.269	306.269	306.269
Other IS	kg/hr	874.545	874.545	874.545	877.207	877.207

The PIPOL for lignin valorization has been designed and projected to be integrated into the biorefinery plant. All other capital costs for the overall biorefinery were assumed to be identical to those for the pretreatment developed based on NREL 2011. In the TEA, a discounted cash flow calculation has been employed to calculate the MPSP, which meets a 10% after-tax internal rate of return. The economics of the lignin bioconversion to PHAs are summarized for the scenarios with different PIPOL technologies from the initial techno-economic evaluation (Table 3). Results indicated that the calculated MPSP will come down as the technologies developed in this project mature over the life of the project. It can be observed that the MPSP for the biorefinery scenario 5 employed SHP-PIPOL was \$11.99/kg, which was highest among these scenarios. Notably, the SHP-PIPOL exhibited the lowest lignin yield, PHA yield, and PHA titer through microbial conversion. The MPSP for AFEX-PIPOL and SEP-PIPOL was \$6.18/kg and \$6.82/kg, respectively, which was lower than other biorefinery designs. These two biorefinery designs also showed higher lignin yield, PHA yield and PHA titer compared with others. The variation tendency of MPSP coincides with

that of annual PHA production from lignin streams, showing that PHA yield and titer are the most crucial factors affecting the MPSP, although total capital cost and the total operating cost varies with every other factor as well. (Page 14)

A further analysis of the unit cost revealed the key cost contributors of MPSP in the PIPOL, when integrated with lignocellulosic biorefinery (Fig. 7). Results showed that the contributors of the MPSP varied with the different biorefinery designs employed. The operating costs (47%~56%), including the utilities, maintenance materials, operating supplies, operating labor, maintenance labor, and control lab labor, etc. were found to be the largest cost contributor per unit of PHAs produced. As one of the major cost contributors in large-scale PHA production, the raw material cost (24%~36%) was the second largest cost contributor for the MPSP. Besides, the annual fixed capital cost was found the third highest cost contributor, following taxes and insurance in the present study. (Page 15)

Response to Reviewer #2 (Remarks to the Author):

Answers to general comments of Reviewer #2:

1: to comment 1: This is a difficult paper to review at least for this journal. I can find little to fault with the paper - it contains a well argued and strongly supported case for altering the unit process design of bio-refineries. The data is clear and persuasive. The article is well written and generously illustrated.

Answer: we appreciate that the reviewer highlighted the results of the manuscript and acknowledged that “our work contains a well argued and strongly supported case for altering the unit process design of bio-refineries. The data is clear and persuasive. The article is well written and generously illustrated”.

We have carried out both experiments and major revision to highlight the key findings of this study. For example, the lignin characterization by 2D-NMR had been done to understand the tailored chemistry by pretreatment and solubilization and their correlations with the lignin reactivity and bioconversion (Supplementary Fig. 5, Page 47). The proposed mechanism had been given in Fig. 8 to show how the integration of pretreatment with ‘plug-in processes of lignin’ (PIPOL) can deconstruct LCC structure of LCB to enhance lignin solubility and reactivity (Fig. 8, Page 41; Supplementary Fig.

8, Page 51). Besides, the overall mass balance around the process have also been provided to highlight the advantage of the new biorefinery design (Supplementary Fig. 7, Pages 49-50). We have also rewritten and refined the manuscript to highlight scientific significance to improve the quality of this manuscript according to reviewers' comments and suggestions.

Fig. 8 Proposed mechanism of the leading pretreatment and 'plug-in processes of lignin (PIPOL)' deconstructed lignin-carbohydrate complex (LCC) structure of lignocellulosic biomass (LCB) to enhance lignin solubility and reactivity (a) and the potential relationships between lignin chemistry and reactivity for bioconversion (b). (Page 41)

Supplementary Fig. 8 Metabolism pathways of lignin and aromatics for the synthesis of polyhydroxyalkanoates (PHAs) through 'biological funnel' in biological lignin valorization by engineering ligninolytic *Pseudomonas putida* KT2440. PIPOL represents 'plug-in processes of lignin' (Page 51)

2: to comment 2: I have no doubt of the value to and potential impact for the designers of bio-refineries but are these likely to read Nature communications? There is also a question over appeal since while the bio-economy is attracting more and more attention and hence more and more research, it is probably still only a narrow section of the science community that is seriously interested in bio-refineries especially from the point of view of unit operations and pre-treatment options of biomass. Overall I think this belongs in a more specialist/technical journal although the editor may think differently.

Answer: We deeply appreciate the reviewer's suggestions. In fact, the comments have helped us to discuss more clearly on the broader impact of the study. In support of the broader impact, we have also carried out more mechanistic study of lignin chemistry and economic impact. After the revision, the current manuscript will appeal a broad audience in environmental sciences (for plastics research), chemistry (for biopolymer chemistry), microbiology, biorefinery, bioeconomy, biomaterial and sustainability.

First of all, the manuscript presents how to tailor the chemistry of a waste stream (lignin) to produce PHA, a bioplastics molecule. One of the four major challenges in sustainability is the consistent accumulation non-degradable material. Petroleum plastics thus imposes a significant challenge for sustainability. Bioplastics holds the promise to address this challenge due to the rapid degradation at the end-life. However, the bioplastics production needs to be cost-effective and sustainable. Using a waste stream to produce PHA-based bioplastics can bring down the cost and promote the sustainability both by replacing petrochemical plastics and by promoting biorefinery waste usage. The manuscript thus addresses profound environmental sustainability challenges and could be interested to audience in biomaterial, environmental sciences and sustainability study.

Second, **based on the reviewer's suggestion, we have expanded the study to discuss mechanisms for tailoring lignin chemistry and improving lignin conversion by microorganisms.** New PIPOL produced more soluble lignin streams with specific reactivities and processibility for microorganisms to consume. We have carried out thorough HSQC NMR analysis to reveal the biological and chemical mechanisms for improved lignin bioconversion. The mechanistic study revealed that PIPOL decreased S/G ratio, β -O-4 and β - β linkage groups, depolymerizing lignin macromolecules to low molecular weight lignin derivatives (Fig. 5, Pages 38; Supplementary Fig. 5, Page 47). PIPOL derived lignin also exposed more phenolic OH groups and COOH groups to enhance the hydrophilicity and hereby the solubility of the lignin (Fig. 6, Page 39). The PIPOL soluble lignin thus has good processibility and better accessibility to ligninolytic *P. putida* KT2440 strains (Fig. 4, Page 37; Fig. 8, Page 41). The findings would guide further design of lignin processing to improve reactivity and processibility. The understanding thus can appeal to broad audience in chemistry, microbiology, polymer sciences, paper and pulping industry.

Fig. 8 Proposed mechanism of the leading pretreatment and 'plug-in' processes of lignin (PIPOL) deconstructed lignin-carbohydrate complex (LCC) structure of lignocellulosic biomass (LCB) to enhance lignin solubility and reactivity (a) and the potential relationships between lignin chemistry and reactivity for bioconversion (b). SEP, steam explosion pretreatment; AFEX, ammonia fiber expansion. (Page 41)

Third, as mentioned by reviewer, the manuscript's impact on bioeconomy is significant. Bio-economy is rapidly growing with a market size of 1.4 trillion in U.S. alone. More importantly, bioeconomy offers the sustainability solutions for the future with broad impacts. What holds up the economic and environmental potential of bioeconomy is the sustainability and profitability of modern biorefineries. A sustainable and economic biorefinery depends on the fractionation of three main polymers in the plant cell wall (cellulose, hemicellulose and lignin) to usable platform molecules to produce a series of value-added products. **PIPOL has a potential to transform lignocellulosic biorefinery and biofuel industry with better economics and sustainability with co-production of multiple products including fermentable sugars for biofuels and lignin for bioplastics.** The initial economic analysis revealed significant potential of the new platform. SEP- and AFEX-integrated PIPOL generated the lower MPSP, corresponding to \$6.82/kg and \$6.18/kg, respectively, compared with other pretreatments (Fig. 7, Page 40). TEA results confirmed that the biological lignin valorization designed as PIPOL minimized the requirement of the additional units,

equipment, and other complicated operations for lignin processing, and thus reduced the total capital cost and the total operating cost of biological lignin valorization to PHAs. Meanwhile, the scenario that incorporates the lignin valorization in a biorefinery could also maximize the synergistic utilization of cellulose, hemicellulose, and lignin, improve the product outputs, amortize the capital investment and hereby promote the industrial implementation of biorefineries.

Overall, the PIPOL together with the mechanistic discovery is significant in guiding the future biorefinery development to improve bioeconomy of biorefinery by maximizing the overall output of biofuel industry. The research is potentially transformative and could have an impact in the fields of biorefining, biomaterials, biofuel, renewable energy, environmental waste utilization, and others.

Response to Reviewer #3 (Remarks to the Author):

Answers to general comments of Reviewer #3:

In this manuscript, new processes were combined and designed as PIP for improving the economy of Biorefinery. However, there are some problems that need to be addressed.

Answer: we appreciate reviewer's suggestion and comment very much for acknowledging the value of the manuscript in improving the economy of biorefinery. We have carried out both experiments and major revision to address these comments. To highlight the findings of this work, the lignin characterization by 2D-NMR had been done to better understand the tailored chemistry by pretreatment and PIPOL (Supplementary Fig. 5, Page 47). Detailed chemical analysis has revealed molecular mechanisms for improved lignin processibility, as shown in Fig. 8 (Page 41). The overall mass balance around the process have also been provided to highlight the advantage of the new biorefinery design (Supplementary Fig. 7, Pages 49-50). The revisions have also been given in the revised manuscript correspondingly.

Fig. 8 Proposed mechanism of the leading pretreatment and 'plug-in processes of lignin (PIPOL)' deconstructed lignin-carbohydrate complex (LCC) structure of lignocellulosic biomass (LCB) to enhance lignin solubility and reactivity (a) and the potential relationships between lignin chemistry and reactivity for bioconversion (b). SEP, steam explosion pretreatment; AFEX, ammonia fiber expansion. (Page 41)

Answers to specific comments of Reviewer #3:

1: to comment 1: plug-in processes seems to be a technology integration, but this concept were not well explained and defined in the manuscript. Please explain the related in the introduction part in detail.

Answer: we appreciate the comment to guide us to improve clarity. We have clearly defined plug-in processes in the revised manuscript and also provided more information in the Introduction, Material and Methods, and Results section in the revised manuscript. In particular, we have changed the name of plug-in process to be more specific as 'plug-in processes of lignin' (PIPOL). The 'Plug-in processes of lignin (PIPOL)' refers to the process of solubilization, conditioning, and fermentation for biological lignin valorization. PIPOL can be integrated with different pretreatment technologies with various configurations to achieve multi-stream integrated biorefinery design. As these operations have been designed to be incorporated into current biorefinery, they were named 'Plug-in processes of lignin (PIPOL)'. By integrating PIPOL with pretreatment,

the biorefinery designs with leading pretreatment technologies were transformed into new scenarios to co-produce multiple products.

Besides presenting a more clear definition, we have also up-graded the Fig. 1 to clearly demonstrated the liquid and solid handling as well as the plug-in concept and PIPOL steps. The revision has also been provided in the revised manuscript according to reviewer’s comments.

Fig. 1 Flow process diagram of the biorefinery scenarios employed leading pretreatments with the incorporation of ‘plug-in processes of lignin (PIPOL)’ (purple module). PIPOL integrates the dissolution, conditioning, and bioconversion of lignin as indicated in different biorefinery scenario as follows. a, a general biorefinery process for ethanol production; b, PIPOL-integrated biorefinery scenario with dilute sulfuric acid pretreatment (DSA); c, PIPOL-integrated biorefinery scenario with steam explosion pretreatment (SEP); d, PIPOL-integrated biorefinery scenario with liquid hot water pretreatment (LHW); e, PIPOL-integrated biorefinery scenario with ammonia fiber expansion (AFEX); f, PIPOL-integrated biorefinery scenario with sodium hydroxide pretreatment (SHP). (Page 34)

The present study aims to address these challenges with the design of a novel ‘Plug-In Processes of lignin (PIPOL)’ for biological lignin valorization that can be directly incorporated into

current biorefinery. The research systemically evaluated lignin bioconversion performance on the integration of PIPOL with five leading pretreatments in biorefinery. The PIPOL had been designed with the integration of solubilization, conditioning and fermentation to improve the solubility and reactivity of lignin and its microbial conversion in biorefinery designs. **(Pages 5 and 6)**

The fundamental challenges for the bioconversion of biorefinery waste lies in the fact that most of the current pretreatment and hydrolysis platforms generate a largely solid lignin waste stream. Our previous studies have established that lignin dissolution is critical for bioconversion, as aromatic monomers and oligomers can serve as substrates for microbes, but not the larger molecules. **Based on this understanding, we have designed a set of procedures, dubbed as ‘plug-in processes of lignin (PIPOL)’, to integrate the solubilization, conditioning, and fermentation for lignin bioconversion. PIPOL is incorporated with current biorefinery to achieve lignin dissolution and bioconversion (Fig. 1a-f).** Considering the features of different leading pretreatments, the PIPOL designs were implemented with various biorefinery configurations and compared in terms of lignin conversion, PHA yield, and economics. **(Page 6)**

In detail, five leading pretreatments that have the potential to be commercially implicated were employed to maximally unleash their roles in the improvements of deconstruction efficiency, lignin processibility, and biorefinery performance. As the soluble low-molecular-weight lignin will enable its microbial conversion 24, 48, the solubilization of the pretreated solid was developed to tailor lignin chemistry and enhance its biological processibility. The soluble lignin was more amenable to microbial conversion into PHA after conditioning. The processing steps of solubilization, conditioning, and fermentation for lignin bioconversion were designed as ‘PIPOL’ to be incorporated into a biorefinery. For acidic pretreatments of DSA and LHW (Fig. 1b and 1d), the lignin solubilization was conducted after separating the pretreated solid in the pretreatment. The soluble lignin stream from the solubilization can be mixed well with the liquid stream from pretreatment to eliminate the conditioning of lignin stream. This PIPOL design integrated the solubilization and fermentation for lignin bioconversion in a biorefinery, where carbohydrates were enzymatically hydrolyzed after the lignin solubilization. As SEP and AFEX were carried out at high solids, the PIPOL design integrated the solubilization, conditioning, and fermentation following the pretreatment (Fig. 1c and 1e). For the alkaline approach, SHP was employed to directly fractionate the lignin from LCB for microbial conversion (Fig. 1f). **(Pages 6 and 7)**

The operations of solubilization, conditioning and fermentation were designed to improve lignin solubility and reactivity for microbial conversion (Fig. 1). As these three steps were directly incorporated into current biorefinery to achieve co-utilization of lignin and carbohydrate at high efficiency. They were named as 'Plug-In Processes of Lignin (PIPOL)'. (Page 20)

2: to comment 2: In figure 5, It seems like the microbe used in this research mainly utilize glucose but not lignin. How to produce ethanol when there were no glucose left?

Answer: we appreciate reviewer's comments to help us to clarify the process. In fact, the PIPOL stream contains primarily lignin and residual sugar. Most of the biorefinery sugar actually remain in the solid portion for hydrolysis and ethanol fermentation, and are not contained in the lignin fraction. The PIPOL processes actually achieve the advantages of improved saccharification efficiency for ethanol production and synergistic utilization of residual sugars and lignin for PHA production. To highlight these points, we have modified Fig. 1 and provided complete mass balance. We hereby expand on the discussion of lignin and sugar usage as follows.

The soluble lignin streams produced from the PIPOL were used as the main carbon source for engineered *P. putida* KT2440 to synthesize PHA (Fig. 1 and 4). Results showed that *P. putida* KT2440 can consume lignin and its derived monomers/oligomers to produce PHAs (Fig. 4, Page 37). The soluble lignin stream fractionated by pretreatment and PIPOL process generally contains lignin and a low level of residual glucose. As shown in Fig. 4, the initial acid insoluble lignin in the liquid stream is 17.2, 16.4, 13.1, 13.9, and 9.2 g/l, respectively, from DSA, SE, LHW, AFEX and SHP with incorporation of PIPOL. About 28.2%, 36.9%, 36.0%, 38.9%, and 34.4% of the acid insoluble lignin has been consumed by *P. putida* KT2440. For acid soluble lignin, about 52.4%, 54.6%, 62.1%, 66.7%, and 51.6% of these lignins have been consumed in fermentation. As compared with previous studies, PIPOL promoted the lignin consumption by *P. putida* KT2440 significantly. In fact, most of the insoluble lignin cannot be consumed by *P. putida* KT2440.

Glucose in the soluble lignin stream is the residual sugar generated from the PIPOL process. Generally speaking, approximately 10-30% of total sugars in biomass

feedstock will be retained in biorefinery residues with most of the traditional pretreatments. These unconverted residual sugars are often highly crystallized, intertwined and embedded with lignin. The chemistry of this LCC (lignin carbohydrate complex) prevents their further hydrolysis. The unprocessed residual sugars not only prevent the lignin processing, but also reduce the overall efficiency of lignocellulosic biomass conversion. Even if they are further released, the sugar concentration in the waste stream will be too low to be utilized alone. Considering all of these factors, the residual sugar negatively impacts the overall economics and reduces the sustainability of biorefineries. However, in the present study, the PIPOL design took into the consideration of co-processing the residual sugars to both promote the lignin bioconversion and improve the overall economics of biorefinery. The concentration of residual glucose was only 5.5, 4.2, 2.8, 2.5, and 3.1 g/l, respectively, from DSA, SE, LHW, AFEX and SHP with incorporation of PIPOL. The residual glucose existed in the soluble lignin stream have been consumed in fermentation. Actually, this residual glucose can be co-consumed in fermentation to promote the bioconversion of lignin to PHAs, and thus also facilitate the utilization of residual glucose, lignin and biomass.

The utilization of the residual sugar does not impact the amount of total hydrolyzed sugar for ethanol fermentation as compared to previous pretreatment studies. Even though solubilization step derived some sugars, most of the carbohydrate remains to be in the solid fraction. Alkaline solubilization actually helped to improve the hydrolysis efficiency of the carbohydrate in biomass.

As shown in Fig. 1, Fig. 3, and Supplementary Fig. 7, the carbohydrates mainly existed in the solid fraction after pretreatment and solubilization, these carbohydrates can be enzymatically hydrolyzed to produce fermentable glucose and xylose. These fermentable sugars could be utilized in fermentation for ethanol production by yeast. In order to clarify the point, we have provided more discussion about the residual glucose in lignin bioconversion in the revised manuscript, along with revised Fig. 1 and mass balance. We really appreciate reviewer's suggestion and comments to help us improve the quality of this manuscript.

Fig. 1 Flow process diagram of the biorefinery scenarios employed leading pretreatments with the incorporation of ‘plug-in processes of lignin (PIPOL)’ (purple module). PIPOL integrates the dissolution, conditioning, and bioconversion of lignin as indicated in different biorefinery scenario as follows. a, a general biorefinery process for ethanol production; b, PIPOL-integrated biorefinery scenario with dilute sulfuric acid pretreatment (DSA); c, PIPOL-integrated biorefinery scenario with steam explosion pretreatment (SEP); d, PIPOL-integrated biorefinery scenario with liquid hot water pretreatment (LHW); e, PIPOL-integrated biorefinery scenario with ammonia fiber expansion (AFEX); f, PIPOL-integrated biorefinery scenario with sodium hydroxide pretreatment (SHP) (Page 34)

Supplementary Fig. 7 Mass balance for the fractionation processes of corn stover biomass by integrating leading pretreatments with the 'plug-in processes of lignin (PIPOL)'. a, PIPOL-integrated biorefinery scenario with dilute sulfuric acid pretreatment (DSA); b, PIPOL-integrated biorefinery scenario with steam explosion pretreatment (SEP); c, PIPOL-integrated biorefinery scenario with liquid hot water pretreatment (LHW); d, PIPOL-integrated biorefinery scenario with

ammonia fiber expansion (AFEX); e, PIPOL-integrated biorefinery scenario with sodium hydroxide pretreatment (SHP). **(Pages 49-50)**

The solubilization in PIPOL produced the soluble lignin stream and generated some residual glucose. As the residual glucose concentration in the soluble lignin stream is too low to be separated and utilized alone, it was co-processed to improve the lignin bioconversion, improving the overall carbohydrate utilization. Although DSA and SHP produced more residual glucose to facilitate the PHA fermentation, AFEX and SEP led to better cell growth, higher PHA concentration and yield, and thus better fermentation efficiency. Taken together, among different pretreatments, AFEX and SEP had the best compatibility with PIPOL to produce more processible lignin. The PIPOL design thus has its unique advantages to improve the biological processibility of lignin and promote its conversion to PHA. **(Page 11)**

3: to comment 3: In figure 6, what exactly is "substrate", is that is glucose or lignin?

Answer: we appreciate reviewer's comments to help us to clarify the manuscript. In the present study, the lignin liquid stream contains mainly lignin and some residual glucose. This lignin stream used as carbon source for PHA production was referred as the substrate. The PHA yield was calculated based on the consumption of lignin and residual glucose by *P. putida* KT2440 during fermentation. Thus, the 'substrate' represents the total amount of lignin and residual glucose consumed by *P. putida* KT2440. The calculation of PHA yield have been given in section of Materials and Methods. We have also provided the detailed information in the revised manuscript.

PHA yield in fermentation was calculated based on the consumption of substrate in medium including both lignin and residual glucose by *P. putida* KT2440. **(Page 23)**

The solubilization in PIPOL produced the soluble lignin stream and generated some residual glucose. As the residual glucose concentration in the soluble lignin stream is too low to be separated and utilized alone, it was co-processed to improve the lignin bioconversion, improving the overall carbohydrate utilization. Although DSA and SHP produced more residual glucose to facilitate the PHA fermentation, AFEX and SEP led to better cell growth, higher PHA concentration and yield, and thus better fermentation efficiency. Taken together, among different pretreatments, AFEX and SEP had the best compatibility with PIPOL to produce more processible lignin. The PIPOL design

thus has its unique advantages to improve the biological processibility of lignin and promote its conversion to PHA. **(Page 11)**

Yours sincerely,

Joshua S. Yuan, Ph.D.

Professor and Director

Synthetic and Systems Biology Innovation Hub

Department of Plant Pathology and Microbiology

Institute for Plant Genomics and Biotechnology

Texas A&M University, College Station, TX 77843

Phone: 979 845 3016

Email: syuan@tamu.edu

REVIEWERS' COMMENTS

Reviewer #1 (Remarks to the Author):

The authors have responded to the comments very thoroughly and the quality of the manuscript has been greatly improved. I do not have any further comments. Nice work!

Reviewer #3 (Remarks to the Author):

In this manuscript, the lignin was used for producing PHA, but the core process, "PIPOL" was still difficult for understanding. I suggest forwarding this article to other journal.